# Revisiting Block-wise Interactions of MMDiT for Training-free Improved Synthesis

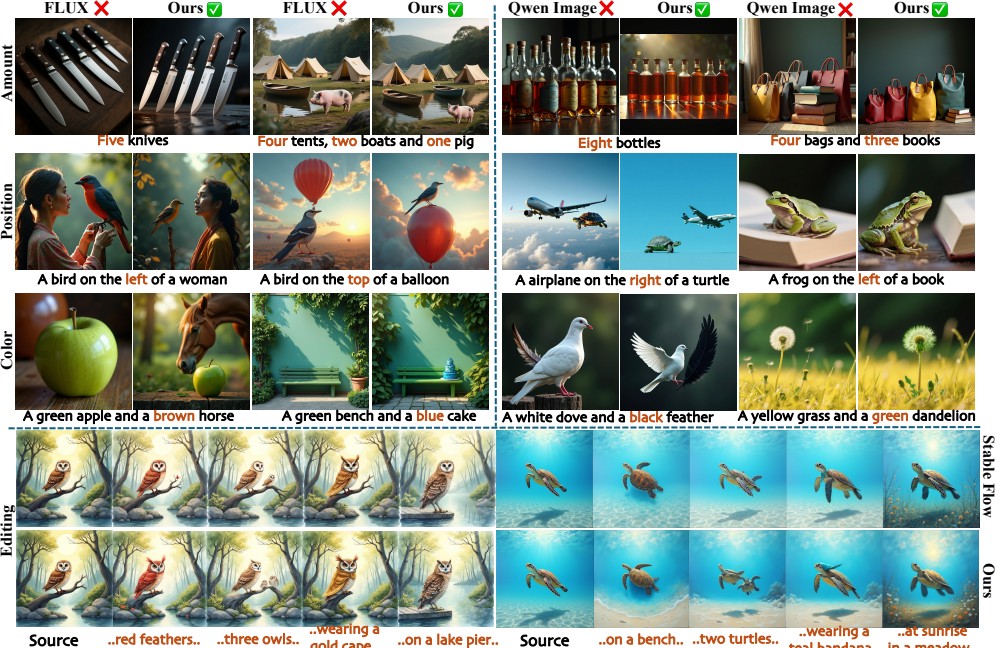

Figure 1: **Visual comparison results on text-to-image generation and image editing** with current state-of-the-art methods, demonstrating improved text alignment across various semantic attributes.

## Abstract

Recent breakthroughs of transformer-based diffusion models, particularly with Multimodal Diffusion Transformers (MMDiT) driven models like FLUX and Qwen Image, have facilitated thrilling experiences in text-to-image generation and editing. To understand the internal mechanism of MMDiT-based models, existing methods tried to analyze the effect of specific components like positional encoding and attention layers. Yet, a comprehensive understanding of how different blocks and their interactions with textual conditions contribute to the synthesis process remains elusive. In this paper, we first develop a systematic pipeline to comprehensively investigate each block's functionality by *removing*, *disabling* and *enhancing* textual hidden-states at corresponding blocks. Our analysis reveals that 1) semantic information appears in earlier blocks and finer details are rendered in later blocks, 2) removing specific blocks is usually less disruptive than disabling text conditions, and 3) enhancing textual conditions in selective blocks improves semantic attributes. Building on these observations, we further propose novel training-free strategies for improved text alignment, precise editing, and acceleration. Extensive experiments demonstrated that our method outperforms various baselines and remains flexible across text-to-image generation, image editing, and inference acceleration. Our method improves T2I-Combench++ from 56.92% to 63.00% and GenEval from 66.42% to 71.63% on SD3.5, without sacrificing synthesis quality. These results advance understanding of MMDiT models and provide valuable insights to unlock new possibilities for further improvements.

## 1 INTRODUCTION

Diffusion models (Ho et al., 2020; Song et al., 2021), especially diffusion transformers (DiT) (Peebles & Xie, 2023; Bao et al., 2023), have become the de-facto paradigm for real-world applications across various domains, including text-to-image (Rombach et al., 2022; Chen et al., 2024) and text-to-video generation (Ma et al., 2024; Yang et al., 2025; Wan et al., 2025), unlocking unprecedented experiences for content creation. In particular, recent breakthroughs such as Stable Diffusion 3 (Esser et al., 2024), FLUX (Labs, 2024), and Qwen-Image (Wu et al., 2025) further advance the synthesis quality via incorporating the flow matching (Lipman et al., 2023; Liu et al., 2023) training objective and the top-performing multimodal diffusion transformer (MMDiT) architecture (Esser et al., 2024; Labs, 2024). Specifically, MMDiT concatenates vision and textual tokens and performs joint self-attention to facilitate a seamless information fusion between these modalities.

Despite MMDiT's remarkable success, it remains unclear how different internal MMDiT blocks interact with textual representations and collaborate with each other to produce coherent outputs. Unlike UNet-based diffusion models (Rombach et al., 2022; Song et al., 2021; Hertz et al., 2022; Zhang et al., 2023) that show a hierarchical coarser to finer semantic representation, MMDiT-based models do not reflect a similar phenomenon due to their isomorphic structure (Peebles & Xie, 2023; Avrahami et al., 2025; Li et al., 2024). Therefore, it is crucial to investigate the intrinsic mechanisms within MMDiT-based models. Several techniques have been proposed to identify the influences of different components and better understand MMDiT. Stable Flow (Avrahami et al., 2025) detected vital blocks by bypassing each block, and TACA (Lv et al., 2025) proposed a timestep-aware attention weighting mechanism to balance multimodal interactions. FreeFlux (Wei et al., 2025) and E-MMDiT (Shin et al., 2025) analyzed MMDiT's attention mechanism by shifting RoPE and decomposing attention metrics, respectively. However, prior studies primarily focus on isolating or manipulating individual aspects, overlooking the synergistic effects that arise from the complex interactions across different blocks and modalities. Consequently, a deeper and detailed analysis of how MMDiT blocks collectively contribute to sophisticated outputs would not only enrich our understanding of MMDiT models but also open avenues for improving synthesis quality and inference efficiency. For instance, by identifying which blocks control specific attributes (*e.g.,* color, amount, spatial relationships), we can revise the corresponding blocks accordingly (see the results in Fig. 1).

To identify each block's detailed role and functionality, this paper conducts a comprehensive analysis of the internal cooperation of MMDiT blocks and their interactions with text conditions. Specifically, we first construct dedicated prompts for each attribute (*i.e.,* color, amount, spatial relationships) and quantify the influence of three popular MMDiT-based models (SD3.5, FLUX, Qwen Image) by: 1) *removing* specific blocks to assess their individual contributions; 2) *disabling* block-level textual conditions to test semantic understanding; and 3) *enhancing* textual hidden-states of different blocks to investigate their potential to refine the coherence and detail of synthesized outputs. Through these analysis, we reveal *several significant findings*: First, semantic information appears in earlier blocks and fine-grained details are rendered in later blocks. Interestingly, different blocks appear to prefer certain semantic attributes, *e.g.,*, earlier blocks handle spatial relations and colors, while relatively later blocks influence amount (as shown the results in Sec. 2). Second, removing blocks is less disruptive than disabling conditions, indicating MMDiT models rely more on conditional guidance and are robust to removing blocks. Last, enhancing textual representations of selective blocks could improve overall text alignment without compromising synthesis quality. These insights clearly clarify the efficacy and interactions of MMDiT components, guiding further optimization and improvements across applications.

Capitalizing on these observations, we develop a novel training-free framework to improve the text alignment, facilitate editing, and accelerate model inference within MMDiT-based models. After identifying each block's contribution to specific semantic attributes, we can strategically enhance their text-visual interactions to improve text alignment. Regarding editing tasks, we can prioritize blocks controlling certain attributes, such as color or amount, ensuring accurate and effective modifications. Additionally, we could accelerate the inference process by skipping blocks that are less critical for semantic understanding, thus streamlining computations while preserving synthesis quality. Together, our framework facilitates efficient, precise, and generalizable model performance across different tasks without requiring additional training. Extensive results show that our method consistently improves performance across various baselines (SD3.5, FLUX, and Qwen Image), evaluation benchmarks (GenEval (Ghosh et al., 2023), T2I-Combench++ (Huang et al., 2025)), metrics

(CLIP Score (Radford et al., 2021)), and different tasks (generation, editing, acceleration), demonstrating its effectiveness and generalizability. More importantly, the overall synthesis quality is maintained at a high standard as evidenced by both automatic metrics (HPSv2 (Wu et al., 2023), Aesthetic Score (Schuhmann, 2022)) and human evaluation.

To sum up, our contributions are: 1) We systematically investigate the internal interactions across blocks and modalities within MMDiT-based models, offering valuable insights to guide further improvements; 2) We develop novel training-free strategies to enhance text-to-image alignment, editing capabilities, and acceleration, fully unlocking the potential of baseline models; 3) Extensive evaluations across multiple baseline models and diverse benchmarks for various tasks consistently demonstrate the effectiveness and generalizability of our approach in advancing model performance.

## 2 SYSTEMATIC ANALYSIS OF BLOCK-WISE INTERACTIONS IN MMDiT

### 2.1 PRELIMINARIES

**Diffusion Models** (DMs) involve a forward process and a reverse generation process. During the forward process, random noise is gradually added to data ($\mathbf{x}_0 \sim q(x)$) across $t \sim (1...T)$ timesteps:

$$\mathbf{x}_t = \sqrt{\alpha_t}\mathbf{x}_{t-1} + \sqrt{1 - \alpha_t}\epsilon_{t-1}. \tag{1}$$

In the reverse generation process, the model iteratively reconstruct the original data following a trajectory opposite to the forward process:

$$p_\theta(\mathbf{x}_{t-1}|\mathbf{x}_t) = \mathcal{N}(\mathbf{x}_{t-1}; \mu_\theta(\mathbf{x}_t, t), \Sigma_\theta(\mathbf{x}_t, t)), \tag{2}$$

where $\mu_\theta$ and $\Sigma_\theta$ are learnable mean and covariance, respectively.

**MMDiT-based Models**, pioneered in SD3 (Esser et al., 2024), leverage a joint multimodal architecture to process text embeddings $\mathbf{c} \in \mathbb{R}^{N_c \times D}$ and visual features $\boldsymbol{x} \in \mathbb{R}^{N_x \times D}$ in a unified attention operation by concatenating them as $h_{in} = [\mathbf{c}; \boldsymbol{x}] \in \mathbb{R}^{(N_c + N_x) \times D}$. This sequence is then processed by multiple MMDiT blocks with a joint self-attention layer:

$$\text{Attention}(Q, K, V) = softmax(QK^T/\sqrt{d_k})V, \tag{3}$$

where $Q, K, V$ denotes the concatenated query, key and value of text and image tokens.

### 2.2 UNDERSTANDING BLOCK-WISE INTERACTIONS OF MMDiT

In this part, we develop a systematic framework to automatically investigate block-wise interactions and their influence on specific semantic attributes (*i.e.,* color, amount, spatial relationships). As shown in Fig. 2, our study involves three key operations: 1) *removing* specific blocks to probe each block's individual importance for generation; 2) *disabling* text conditions of different blocks to evaluate their reliance on textual guidance; 3) *enhancing* textual representations of certain blocks to investigate their potential to refine both coherence and detail in synthesized outputs. Specifically, we amplify the text condition hidden states by a factor of 2 as $\mathbf{c} \rightarrow 2\mathbf{c}$, to investigate each block's latent capacity for assimilating semantic information. Regarding the disabling operation, we mute the textual hidden states via attaching an empty tensor with $torch.zeros\_like(c)$.

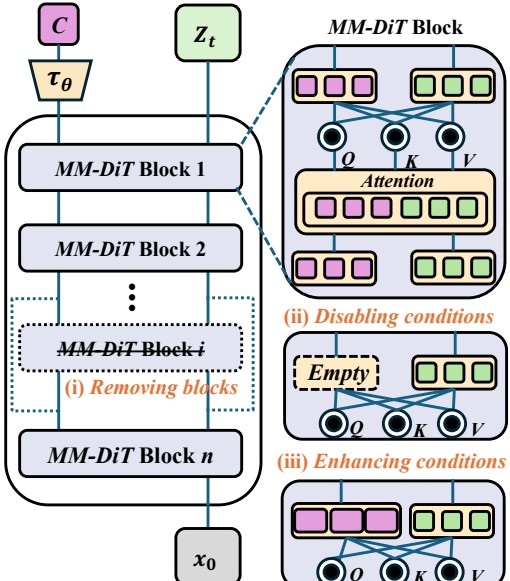

Figure 2: **Systematic analysis overview.**

Then, we construct a challenging prompt dataset with GPT-5, comprising 333 diverse and difficult prompts across three attributes: color, amount, and spatial relationships. For each prompt, we

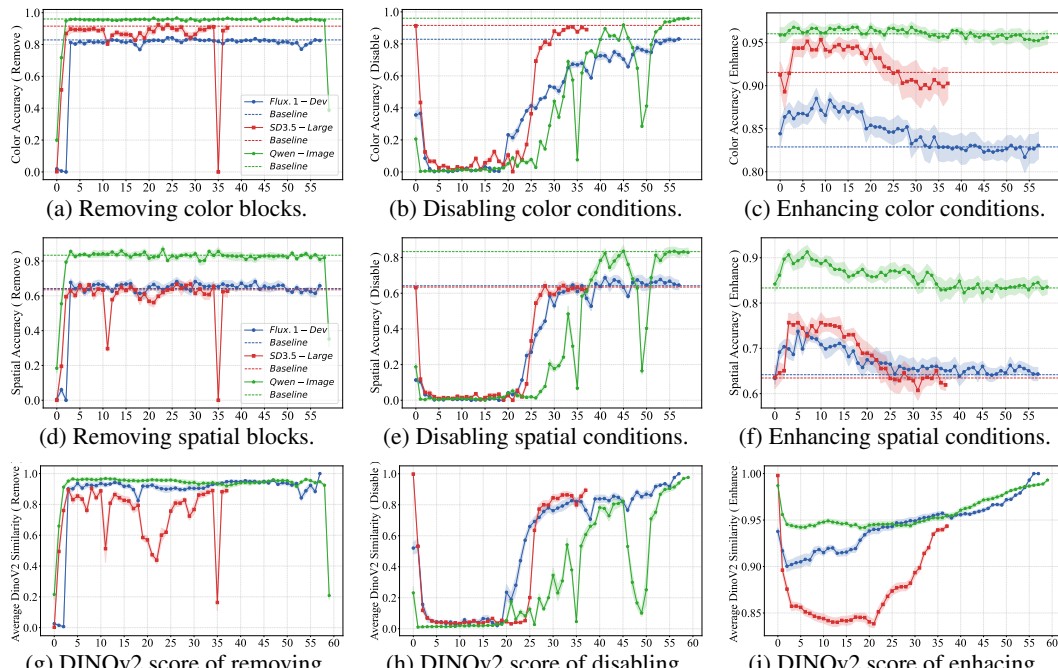

Figure 3: **Block-wise analysis results across various MMDiT-based models on different attributes**. We identify each block's specific role in generating images and its interactions with textual conditions. The accuracy of different attributes is evaluated by QwenVL-2.5-72B on multiple runs, and DINOv2 score shows the perceptual similarities. See supplementary Fig. A2 for more results.

perform *removing*, *disabling* and *enhancing* on SD3.5-Large (Stability-AI, 2024), FLUX.1-Dev (Labs, 2024), and Qwen Image (Wu et al., 2025) models, operating on one block at a time. Finally, for color and spatial relationships, we evaluate generated images using Qwen2.5-VL-72B (Bai et al., 2025) via question-answering pairs on the prompts and the generated images. Regarding the amount attribute, we adopt CountGD (Amini-Naieni et al., 2024) to precisely evaluate the numeracy results. Further, we evaluate perceptual (DINOv2 (Oquab et al., 2023)) and semantic similarities (CLIP Score (Radford et al., 2021)) between images from our modified and original models to quantify the effect of our block-wise manipulations. Notably, each evaluation is averaged over five runs with different random seeds to ensure reliability (see supplementary Sec. D.1 for more details).

The analysis results on different attributes across SD3.5 (38 blocks), FLUX (58 blocks), and Qwen Image (60 blocks) are shown in Fig. 3. For each subfigure, we plot the quantitative curves of performing our analysis method, *i.e.,* removing (1st column), disabling (2nd column), enhancing (3rd column), on three semantic attributes, namely color, spatial relationships and amount. Despite testing on different models, we could consistently observe several interesting findings from these results.

**Removing less critical blocks tends not to significantly impact overall performance.** Fig. 3a and 3d illustrate the impact of *removing* different blocks. We could observe that all models are sensitive to removing earlier $(0 - 5)$ and late blocks, causing significant performance drops. We attribute this sensitivity to the critical roles of early blocks in initializing inputs and of late blocks in refining details for the final output. By contrast, removing middle-layer blocks generally has a smaller impact on synthesis performance and DINOv2/CLIP scores. Such observation indicates that these blocks might be less critical for maintaining the fidelity and coherence of the generated outputs. Thus, some of these blocks can be removed to improve efficiency without degrading quality.

**Disabling textual conditions is more disruptive than removing specific blocks.** Fig. 3b and 3e reveal that disabling textual conditions, especially in the earlier blocks $(0 - 20)$, causes a more pronounced degradation in the synthesis performance compared to merely removing specific blocks. That is, textual conditions paly a crucial role in guiding the models' generative process. Moreover, we can see that the results of disabling conditions of late blocks are less detrimental to the overall performance, particularly the CLIP Score, suggesting that these blocks are specialized in refining details and the core semantics are rendered by the earlier blocks.

**Enhancing textual conditions on certain blocks could improve the synthesis performance.**
Fig. 3c and 3f show the enhancing results. Though the simple ×2 operation may not yield optimal results, enhancing textual conditions on certain blocks can improve the synthesis performance. Remarkably, for color and spatial attributes, all models show performance improvements compared with the original baseline, despite Qwen Image showing less improvement due to its strong baseline. Interestingly, different blocks seem to reflect a preference for certain semantic attributes, *e.g.,* earlier blocks improve color and spatial attributes, while enhancing later blocks benefits amount. To our knowledge, this observation has never been documented in existing literature. In return, one could manipulate specific attributes (e.g., amount or color in Fig. 1 and 6) by altering textual information at the corresponding blocks. Additionally, enhancing textual conditions by ×2 can sometimes reduce performance (SD3.5 amount, Fig. A2c). This may result from the enhancements exceeding the model's activation range or targeting incorrect blocks. (See Sec. 4 and D.3 for detailed results.)

Overall, our analysis provides a comprehensive investigation of the block-wise capabilities and their interactions with textual conditions, yielding several interesting insights on how different blocks contribute to the output. These findings contribute to a better understanding of MMDiT-based models, offering valuable perspectives that could facilitate further enhancements and optimizations.

## 3 METHODOLOGY

Based on the insights derived from our systematic analysis of block-wise interactions in MMDiT models, we propose innovative, training-free techniques designed to boost performance. After assessing each block's contribution to semantic understanding and the final output, we propose to 1) enhance textual-visual interactions in pivotal blocks, 2) precisely edit specific attributes in blocks that dominantly control them, and 3) accelerate generation by removing low-impact blocks.

### 3.1 ENHANCING TEXT-VISUAL INTERACTIONS

We propose a straightforward, training-free method to enhance text-visual interactions within blocks by capitalizing on their pivotal roles. Specifically, we enhance the hidden states of textual conditions in these vital blocks $\mathcal{V}$ by a factor of $\lambda(l)$:

$$\boldsymbol{c}_{enh}^{(l)} = \lambda(l) \cdot \boldsymbol{c}^{(l)}, \quad \forall l \in \mathcal{V}, \tag{4}$$

where $\boldsymbol{c}^{(l)}$ denotes the original textual hidden states of block $l$ and $\odot$ is element-wise multiplication. $\lambda(l)$ can be a constant or a block-dependent function.

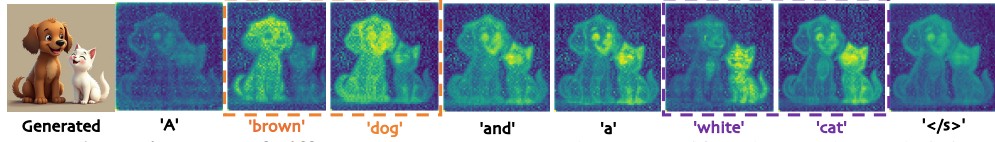

Figure 4: **Attention map of different tokens**. We can enhance specific tokens to boost their impact.

**Token-level Enhancement.** To further improve the semantic understanding capability of certain blocks on specific attributes, we introduce token-level enhancement to amplify key textual tokens. As shown in Fig. 4, such an operation ensures that critical semantic attributes receive greater emphasis. Formally, let $M$ denote the corresponding indices of enhanced textual tokens, we perform:

$$\boldsymbol{c}_{enh}^{(l)} = (1 - M) \odot \boldsymbol{c}^{(l)} + \lambda(l) \cdot M \odot \boldsymbol{c}^{(l)}, \quad \forall l \in \mathcal{V}. \tag{5}$$

Then, the enhanced textual signals $\boldsymbol{c}_{enh}^{(l)}$ are then concatenated with vision signals: $h_{in} = [\boldsymbol{x}^{(l)}, \boldsymbol{c}_{enh}^{(l)}]$ as the input of following blocks. In this way, our method allows for a better understanding of textual conditions, emphasizing key semantic attributes within the model.

### 3.2 ENABLING PRECISE TEXT-BASED EDITING

We incorporate our enhancement into editing tasks, facilitating precise textual editing with the target text instructions. Specifically, we perform image editing via parallel generation following Avrahami et al. (2025), producing the source image $I$ and target image $\hat{I}$ in parallel from the source prompt $p_{\text{src}}$ and edited prompt $p_{\text{tgt}}$. During inference, self-attention features from the source image are injected into the target image to preserve visual content. However, existing methods often struggle

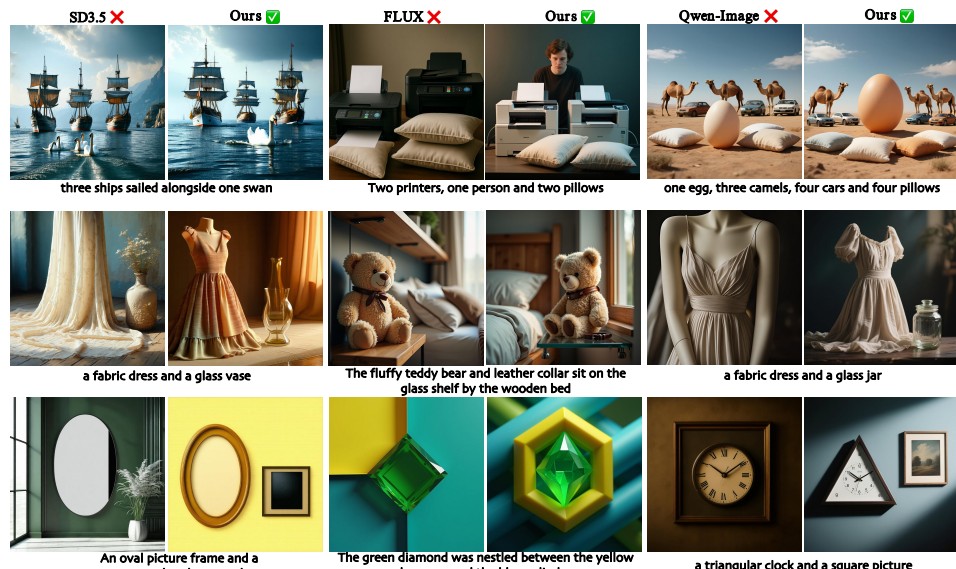

SD3.5 ❌  Ours ✅  FLUX ❌  Ours ✅  Qwen-Image ❌  Ours ✅

three ships sailed alongside one swan  Two printers, one person and two pillows  one egg, three camels, four cars and four pillows

a fabric dress and a glass vase  The fluffy teddy bear and leather collar sit on the glass shelf by the wooden bed  a fabric dress and a glass jar

An oval picture frame and a rectangular photograph  The green diamond was nestled between the yellow hexagon and the blue cylinder  a triangular clock and a square picture

Figure 5: **Qualitative comparisons between baselines and our method**. Our method significantly improves the text alignment across various semantic attributes including amount, colors, textures, and complex prompts, *etc.* Zoom in for details and see more results in the supplementary Sec. F.

to precisely edit specific attributes due to limited alignment with target instructions. Our empirical findings motivate us to enhance target prompts $\hat{p}$ across critical blocks to improve the editing accuracy. Formally, the self-attention injection is performed as:

$$K_t^{tgt,(l)} \leftarrow [K_t^{I_{src},(l)}; K_t^{p_{tgt}^{enh},(l)}], \quad V_t^{tgt,(l)} \leftarrow [V_t^{I_{src},(l)}; V_t^{p_{tgt}^{enh},(l)}], \quad \forall l \in \mathcal{V}$$
$$O_t^{(l)} = softmax(Q_t^{(l)}(K_t^{tgt,(l)})^T/\sqrt{d})V_t^{tgt,(l)}, \tag{6}$$

where $p_{tgt}^{enh}$ denotes the enhanced target text embeddings using Eq. 5. Such enhancement enables the model to concentrate on the attributes indicated by target text instructions, thereby improving the editing accuracy as shown in Fig. 1 and 6.

### 3.3 Accelerating Inference Process

Recall that our analysis indicates that removing some blocks causes a smaller impact on the output, suggesting their role in rendering fine-grained details instead of vital semantics.

Accordingly, we accelerate inference with a training-free mechanism by skipping specific blocks identified as less critical from our probing analysis, denoted as $\mathcal{S} = \{s_1, s_2, \ldots, s_m\}$. Then, for a skipped block $s$, the input feature for the next block is:

$$Z_{out}^{(s)} = Z_{out}^{(s-1)} \text{ if } (s \in \mathcal{S}), \quad \text{Block}^{(s)}(Z_{out}^{(s-1)}) \text{ otherwise } (s \notin \mathcal{S}). \tag{7}$$

Notably, when classifier-free guidance (CFG) is enabled during inference, we can achieve a $\times 2$ inference time acceleration on both the conditional and unconditional predictions.

## 4 Experiments

### 4.1 Implementation Details

**Baseline Models.** We apply our method on several state-of-the-art MMDiT-based models, namely SD3.5-Large (Stability-AI, 2024), FLUX.1-Dev (Labs, 2024) and Qwen Image (Wu et al., 2025). Based on the observations from our systematic analysis in Sec. 2, we perform our enhancement on selected pivotal blocks according to the attribute in Tab. 1. We evaluate the editing and acceleration performance on FLUX.1-Dev, the editing instructions are enhanced based on their semantic attributes. For acceleration, we remove blocks that are identified less critical for the final output, *i.e.,* blocks in $20 - 40$ of FLUX. Additionally, we also incorporate our method into TeaCache (Liu et al., 2025) to testify our compatibility. Regarding the comparison methods, we compare our method on T2I generation with TACA (Lv et al., 2025), which attach a timestep-aware importance on the

Table 1: **Selected blocks for enhancing, editing, and acceleration for different attributes**.

| Models | Total Blocks | Color | Spatial | Amount | Others (Shape, *etc.*) |
|---|---|---|---|---|---|
| SD3.5-Large Stability-AI (2024) | [0, 37] | [3,9,15,20] | [3,10,17,22] | [26,29,33,36] | [3,9,15,21] |
| FLUX.1-Dev Labs (2024) | [0, 57] | [2,8,14,20,28] | [2,7,14,20,27] | [32,37,45,49,54] | [2,7,12,17,22] |
| Qwen Image Wu et al. (2025) | [0, 59] | [4,11,17,24,29] | [3,8,11,19,28] | [34,40,45,51,54] | [3,9,15,21,27] |

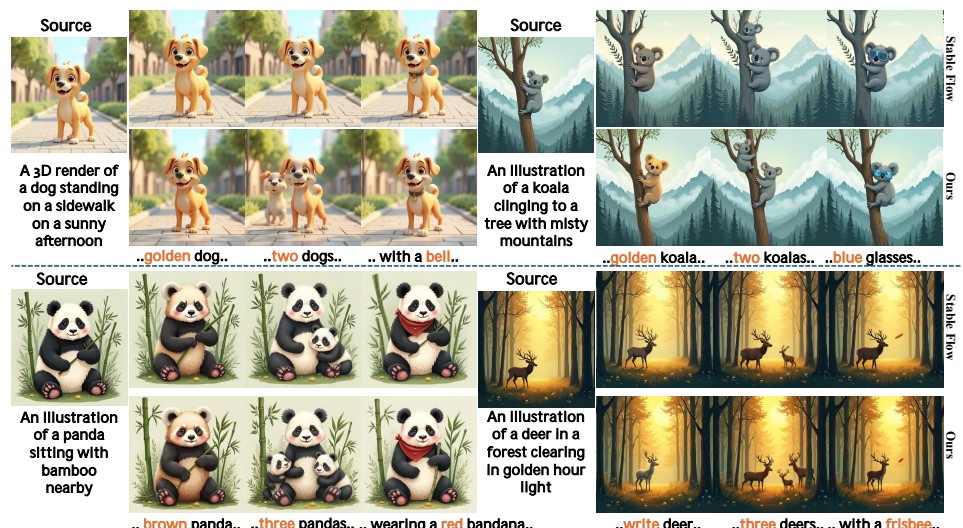

Figure 6: **Qualitative comparison of editing results between Stable Flow and our method**. Our method enables more precise editing on specific attributes on changing the color, amount, *etc.*

textual conditions within the attention. We implement our method on Stable Flow (Avrahami et al., 2025) and compare with it, leaving other details untouched. The enhancing parameter $\lambda(l)$ in Eq. 5 is set to $1.5$ unless otherwise specified. All inference parameters, including CFG scale, denoising steps, *etc.*, follow the official default settings for our systematic analysis in Sec. 2 and the evaluation results. All experiments are carried out on NVIDIA 4090 and H100 GPUs.

**Datasets and Evaluation metrics.** We evaluate our method on the T2I-CompBench++ (Huang et al., 2025) and GenEval (Ghosh et al., 2023) benchmarks, which are widely adopted for text-to-image alignment. We follow the official guidance for evaluation. For instruction-based editing, we employ GPT-5 to generate $1,000$ diverse source–target text pairs, each with multiple editing instructions (*e.g.*, color change, object addition), yielding $4,000$ evaluation samples. We adopt $\text{CLIP}_{img}$ to evaluate the similarity between the source images and edited images, and $\text{CLIP}_{txt}$ (Radford et al., 2021) score to evaluate the alignment between edit instructions and the edited images. Additionally, we utilize Aesthetics score (Schuhmann, 2022) and HPSv2 (Wu et al., 2023) to evaluate the overall image quality, ensuring that our method does not negatively impact the synthesis quality of the original models. Furthermore, we perform human evaluations on text-to-image and image editing tasks, 3 individuals are presented with random $100$ images and asked to select images with higher quality. More details are presented in supplementary Sec. E.

### 4.2 MAIN RESULTS

**Improved Text Alignment of Text-to-Image Generation.** Tab. 2 and 3 shows the quantitative results on T2I-CompBench++ and GenEval benchmarks. We could observe that our proposed method consistently obtains performance gains across various attributes on all three models, demonstrating the superiority and flexibility of our method. Remarkably, we achieve substantial improvement of 12% on Shape, 10% on Texture, and 8% on Color, in a totally training-free manner. Additionally, the quantitative results of HPSv2 and Aesthetics scores demonstrate that our method improves the text alignment while maintaining the high aesthetic quality. Together with the quantitative results, the qualitative results in Fig. 1 and 5 further show the efficacy of our method on improving semantic understanding across various attributes.

**Instruction-based Editing Results.** The quantitative comparison results of our method and the baseline Stable Flow (Avrahami et al., 2025) are presented in Tab. 4. Our method outperforms Stable Flow on $\text{CLIP}_{txt}$ score (↑0.94), showing more accurate editing towards textural instructions.

Table 2: **Quantitative results on T2I-CompBench++**. * denotes token-level enhancement.

| Model | Attribute Binding | | | Object Relationship | | | Amount* | Complex | Image Quality | |
|---|---|---|---|---|---|---|---|---|---|---|
| | Color | Shape | Texture | 2D Spatial | 3D Spatial | Non-Spatial | | | HPSv2 | Aes. |
| TACA | 0.7434 | 0.5784 | 0.7444 | 0.2947 | 0.3839 | 0.3114 | 0.6029 | 0.3820 | **29.3225** | **6.2297** |
| SD3.5 | 0.7284 | 0.5592 | 0.7471 | 0.2866 | 0.3816 | 0.3118 | 0.5969 | 0.3727 | 29.2869 | 6.0978 |
| + Ours | **0.8052** | **0.6744** | **0.8428** | **0.3647** | **0.3923** | **0.3169** | **0.6088** | **0.4047** | 28.9501 | 5.9401 |
| TACA(r=64) | 0.7535 | 0.5126 | 0.6522 | 0.3043 | 0.3814 | 0.3045 | 0.5855 | 0.3619 | 29.1525 | 6.3327 |
| TACA(r=16) | 0.7296 | 0.4898 | 0.6549 | 0.2991 | 0.3790 | 0.3034 | 0.5780 | 0.3585 | 29.1375 | 6.3205 |
| FLUX | 0.7322 | 0.4908 | 0.6490 | 0.2935 | 0.3739 | 0.3044 | 0.5877 | 0.3597 | 29.1586 | 6.3563 |
| + Ours | **0.7804** | **0.5482** | **0.6980** | **0.3280** | **0.3900** | **0.3054** | **0.6091** | **0.3691** | **29.2267** | **6.4110** |
| Qwen Image | 0.8554 | **0.6358** | 0.7650 | 0.3973 | 0.4077 | 0.3110 | 0.7406 | 0.3983 | 28.8831 | 6.1925 |
| + Ours | **0.8677** | 0.6348 | **0.7796** | **0.4560** | **0.4202** | **0.3123** | **0.7616** | **0.4104** | **29.0212** | **6.2378** |

Table 3: **Quantitative results on GenEval**. * denotes token-level enhancement.

| Model | Overall | Single object | Two object | Counting* | Colors | Position | Color attribution | HPSv2 | Aes. |
|---|---|---|---|---|---|---|---|---|---|
| SD3.5 | 0.6642 | 0.9438 | 0.8939 | 0.6344 | 0.8059 | 0.2325 | 0.4750 | **29.5759** | **5.8871** |
| + Ours | **0.7163** | **0.9781** | **0.9672** | **0.6375** | **0.8650** | **0.3925** | **0.4825** | 29.3729 | 5.7902 |
| FLUX | 0.6538 | **0.9904** | 0.8258 | 0.6375 | 0.7713 | 0.2575 | 0.4400 | 29.8115 | 6.3650 |
| + Ours | **0.6826** | 0.9688 | **0.8914** | **0.6438** | **0.7739** | **0.3475** | **0.4700** | **29.8207** | **6.4043** |
| Qwen Image | 0.8551 | **0.9906** | 0.9520 | 0.8562 | 0.8617 | 0.7375 | 0.7325 | 30.4510 | **6.2327** |
| + Ours | **0.8777** | **0.9906** | **0.9722** | **0.8594** | **0.8989** | **0.7475** | **0.7975** | **30.6851** | 6.2113 |

Meanwhile, the $\text{CLIP}_{img}$ similarity remains nearly unchanged ($\downarrow$0.008), suggesting that our method effectively enables more precise editing in line with the given instructions while preserving the visual integrity and coherence of the images. Furthermore, the result of human preference further reflects the effectiveness of our method. Combined with the qualitative results in Fig. 1 and 6, these results highlight also the efficacy of our method.

**Inference Acceleration.** Tab. 5 reports inference acceleration results by skipping less critical blocks, showing averaged inference time over 400 prompts on NVIDIA 4090 and H100 GPUs. The results show that our method substantially reduces inference time and can be seamlessly com-

Table 4: **Image editing results**.

| Method | $\text{CLIP}_{img}$ | $\text{CLIP}_{txt}$ | Human Preference |
|---|---|---|---|
| Stable Flow | **0.9642** | 35.2584 | 40.83% |
| + Ours | 0.9637 | **36.1988** | **59.17%** |

bined with existing acceleration techniques Liu et al. (2025) for further acceleration. Importantly, image quality metrics (*i.e.,* HPSv2, Aesthetic, $\text{CLIP}_{txt}$) confirm that synthesis quality is preserved with accelerated inference.

### 4.3 ABLATION ANALYSIS

**Analysis on the scale of $\lambda(l)$.** Here, we investigate the sensitivity of the scale $\lambda(l)$ to identify its impact. Specifically, we evaluate the performance of different attributes on FLUX with $\lambda(l)$ ranging from 1.2 to 2.0. As shown in Fig 7 (a), our method consistently achieves significantly better results than the baseline despite some fluctuations, indicating the effectiveness of our method. Additionally, we also evaluate the performance of weakening the textual conditions in Tab. 6. It turns out that the weakening operation significantly decreases the model's performance, further demonstrating the importance of these vital blocks and validating the soundness of our method.

**Analysis on the selection of enhanced blocks.** To evaluate the effectiveness of our analysis in selecting the proper number of blocks for enhancement, we apply our enhancement to varying block counts $N \in \{1, 3, 5, 7, 9\}$. Fig. 7(b) shows that increasing $N$ initially boosts performance, but manipulating more blocks ($> 9$) might lead to degradation due to distribution shift. Furthermore, we perform enhancement on random chosen blocks of FLUX

Table 5: **Acceleration results**.

| Method | Time(4090) | Time(H100) | HPSV2 | Aes |
|---|---|---|---|---|
| FLUX | 36.7889$s$ | 13.0876$s$ | **29.0533** | **6.1903** |
| + Ours | **31.6931$s$** | **11.3010$s$** | 28.8408 | 6.1034 |
| TeaCache | 26.6187$s$ | 9.6125$s$ | **28.8951** | **6.2067** |
| + Ours | **24.5276$s$** | **8.8804$s$** | 28.8647 | 6.1801 |

(5 and *all*) rather than our selected blocks, the results are given in Tab. 6. We can derive from the table that enhancing randomly selected or all blocks underperforms enhancing dedicated blocks identified from our analysis, highlighting the efficacy of our proposed approach. What's more, this observation also reflects that different blocks do not contribute equally to different attributes, consistent with our findings in Sec. 2. More analysis results are given in the supplementary Sec. F.

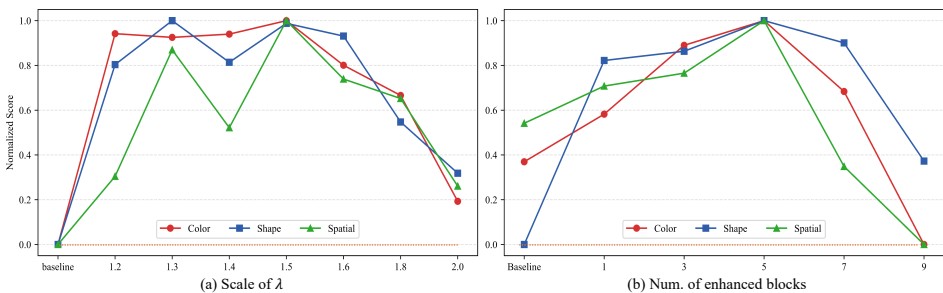

Figure 7: **Ablation analysis on the enhancing scale (*left*) and the selection of blocks (*right*).**

## 5 RELATED WORK

**Diffusion Transformers** DiT (Peebles & Xie, 2023) have become the dominant paradigm for high-fidelity image and video generation, which adopt transformer (Vaswani et al., 2017) architecture as the main backbone, demonstrating superior scalability and training efficiency compared to previous UNet-based (Ho et al., 2020; Dhariwal & Nichol, 2021; Ho & Salimans, 2022) models. Recent variants, such as open-sourced SD3 (Esser et al., 2024), FLUX (Labs, 2024), Qwen Image (Wu et al., 2025), Hunyuan Image (Team, 2025b) and Hunyuan Video (Team, 2024), and commercial models like

Table 6: Ablation analysis on smaller $\lambda$ and block selections.

| Methods | Color | Shape | 2D Spatial |
|---|---|---|---|
| 0.7 | 0.3161 | 0.2653 | 0.1100 |
| 0.9 | 0.6891 | 0.4395 | 0.2611 |
| Random 5 blocks | 0.7624 | 0.5072 | 0.3119 |
| *All* blocks | 0.2360 | 0.2736 | 0.0495 |
| Ours | **0.7804** | **0.5482** | **0.3280** |

Seedream series (Gong et al., 2025; Gao et al., 2025), Sora (OpenAI, 2024), Imagen3 (Baldridge et al., 2024), further advance text-to-image/video to an unprecedented level with the top-performing multimodal diffusion transformer (MMDiT) architecture. Besides scaling the MMDiT-based models, many efforts have also focused on accelerating the iterative denoising process (Lu et al., 2022; Song et al., 2023; Luo et al., 2023), controlling the results (Zhang et al., 2023; Tan et al., 2024; Xiang et al., 2025), editing the outputs (Avrahami et al., 2025; Wei et al., 2025), *etc.*

**Understanding and Improving Diffusion Models.** Numerous prior works proposed various techniques to analyze the roles of different components of UNet-based diffusion models. For instance, P2P (Hertz et al., 2022) showed that cross-attention layers are essential for rendering the spatial layout, MasaCtrl (Cao et al., 2023) and Liu et al. (2024a) demonstrated that self-attention maps are more important for preserving the geometric and shape details. FreeU (Si et al., 2024) and PBC (Zhou et al., 2025) respectively analyzed the functionality of skip connections and position encoding mechanism in diffusion UNet. Further, Yi et al. (2024) investigated the working mechanism of text prompts and Williams et al. (2023) developed a unified framework for designing and analysing UNet architectures. However, the understanding of MMDiT components remains under-explored, and it is crucial to gain a comprehensive insight into these components to advance the field. Existing approaches explored the roles of layers (Avrahami et al., 2025), rotary position embeddings (RoPE) (Wei et al., 2025), and attention embeddings (Shin et al., 2025), but often focus on specific applications like editing and lack systematic evaluation of MMDiT components. TACA (Lv et al., 2025) indicated an imbalanced issue in the cross-model attention and ameliorated this with a timestep-aware weighting scheme. Nevertheless, none of the current approaches provides a holistic view of how these components jointly influence the model's overall performance and versatility.

## 6 CONCLUSIONS

**Conclusions.** In this work, we systematically analyze block-wise contributions and their interactions with text conditions, offering a better understanding of the internal mechanisms within MMDiT-based generative models. Meanwhile, our analysis reveals several valuable findings that unlock new possibilities for improving the synthesis quality. Based on these findings, we propose training-free techniques for improved text alignment, precise semantic editing, and accelerated inference. Extensive results demonstrate the effectiveness of our method.

**Limitations and Future Works.** Despite substantial performance gains, our method has limitations. It relies on automatic block-wise analysis and cannot perfectly synthesize highly complex prompts due to pretraining constraints. Future work could incorporate trainable modules and token-level dynamic routing to further improve synthesis quality and semantic understanding.

**Ethics Statement.** This work follows the ICLR Code of Ethics. Experiments use only public datasets and models, and our constructed dataset will be released. No sensitive or personally identifiable data is involved.

**Reproducibility Statement.** All implementation details, hyperparameters, and evaluation protocols are provided in the paper and appendix. We will release code and scripts upon publication to ensure reproducibility: `https://anonymous.4open.science/r/Revisiting_MMDiT_for_Improved_Synthesis/readme.md`.

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

CONTENTS OF APPENDIX

## A  Overview of Appendix

The appendix provides supplementary information and results supporting the main text. It begins with a discussion on the use of large language models (LLMs) and Limitations This is followed by a detailed analysis of probing results (Sec. D), including descriptions of baseline models and implementation details, datasets used for probing, quantitative metrics such as amount and CLIP score analysis, and evaluation procedures. Next, the appendix (Sec. E) presents the detailed setup of evaluation results, covering inference configurations, evaluation benchmarks, token-level enhancement mask localization, editing task details, and acceleration experiments. Additional quantitative results (Sec. F) are provided, including analyses on enhanced block selection, scaling schemes, ablation studies, and token-level enhancements. More qualitative results (Sec. G) show visual examples across different models and tasks, such as generation, editing, and model-specific outputs. The appendix also discusses failure cases and limitations observed during experiments. Last, the human evaluation protocol used to assess output quality and alignment (Sec. H). Together, these sections offer a comprehensive resource for reproducing experiments, understanding model behavior, and exploring additional findings beyond the main text.

## B  Use of Large Language Models (LLMs)

We made limited use of Large Language Models (LLMs) in preparing this work, specifically for the following purposes. **(1)**.Writing assistance. The LLMs were used to polish grammar and phrasing in parts of the paper. But all substantive content, ideas, and claims were written by the authors. **(2)**.Prompt generation for probing analysis and editing prompts. GPT-5[1] was used to generate an initial set of candidate prompts for probing analysis and editing prompts. The final prompts were manually screened, refined, and verified by the authors before use. All research contributions, analyses, and conclusions are solely the responsibility of the authors.

## C  Limitations

Although our proposed method delivers substantial performance gains across multiple tasks, certain limitations remain.

**Dependence on the preliminary block-wise analysis.** A core part of our enhancement pipeline relies on an initial, automatic analysis that identifies block-wise interactions inside the model. Luckily, our analysis is totally automatic and could be performed across various models with different numbers of blocks, varied model sizes, as well as different models (e.g., SD3.5, FLUX, Qwen Image).

**Limited fidelity on very complex prompts.** For extremely complex or highly detailed prompts, our method may fail to capture all fine-grained elements. This limitation mainly stems from the pretraining data distribution, where rare object combinations or subtle high-frequency details are underrepresented, sometimes resulting in missing elements or artifacts in the output. As could also be observed from the failure cases in Fig. A9.

**Evaluation and generalization limits.** Our evaluation on standard benchmarks may not fully reflect perceptual quality or robustness, and gains might not generalize to niche domains or prompts requiring world knowledge absent from pretraining.

---

[1]https://chatgpt.com

# D DETAILED ANALYSIS OF PROBING RESULTS

## D.1 BASELINE MODELS AND IMPLEMENTATION DETAILS

We implement our probing analysis based on the widely adopted MMDiT-based text-to-image models, including:

- Stable Diffusion 3.5-Large[2] (Stability-AI, 2024): A latent diffusion model with approximately 8 billion parameters, based on the Multimodal Diffusion Transformer (MMDiT) architecture. It demonstrates strong performance in prompt adherence, typography, and supports a mature ecosystem of extensions.
- FLUX.1-Dev[3] (Labs, 2024): A 12B-parameter rectified-flow transformer model that adopts advanced training techniques and a substantially larger dataset to enhance visual fidelity. It has attracted significant community attention for its improvements in prompt alignment, detail rendering, and efficient sampling.
- Qwen Image[4] (Wu et al., 2025): A 20B-parameter MMDiT model developed within the Qwen series, designed for robust multimodal reasoning and high-quality image synthesis. It is particularly noted for its strong performance in complex text rendering (especially Chinese) and text-guided image editing.

We use the official checkpoints provided by the authors and the *diffusers* library (Team, 2025a) for implementation. During inference, model weights are loaded in 16-bit precision. No acceleration techniques such as xformers or memory-efficient attention are used. The default parameters during inference are summarized in Tab. A1.

Table A1: Model information and default parameters during inference.

| Models | SD3.5-large | FLUX.1-Dev | Qwen Image |
|---|---|---|---|
| MMDiT Blocks | [0,37] | [0,57] | [0,59] |
| Parameters | 8B | 12B | 20B |
| Inference Steps | 28 | 28 | 50 |
| CFG Scale | 7.0 | 3.5 | 4.0 |
| Size | (1024,1024) | (1024,1024) | (1024,1024) |

During probing analysis, we use identical hyperparameters for all models to ensure fair comparison, as summarized in Tab. A1. For each model with $N$ blocks, we fix a random seed and generate one baseline image, $N$ *disable* images, $N$ *remove* images, and $N$ *enhance* images—constituting one experimental group. For each of the three MMDiT-based models, we conduct five experimental groups using five different random seeds $(0, 42, 329, 1234, 99514)$. Final results are reported as the average across these five groups.

## D.2 CONSTRUCTED DATASET FOR PROBING ANALYSIS

We construct a challenging prompt dataset with GPT-5, comprising 333 diverse and complex prompts across three attributes: color (129 prompts), spatial relationship (104 prompts), and amount (100 prompts). For color attributes, we focus on objects with distinctive colors (e.g., "red apple", "yellow banana"). For spatial relationships, we include prompts describing eight positional relations (e.g., "left", "right", "above", "below", "upper left", "upper right", "lower left", "lower right"). For amount attributes, we cover a range of quantities from "three" to "nine". We also ensure diversity in object categories, including human, animal, natural scenes, indoor scenes, food, clothing & accessories, vehicles, and so on.

The distribution of prompts across these attributes is illustrated in Fig. A1. We ensure that the prompts are diverse and challenging, covering various object categories, colors, spatial relations, and quantities.

---

[2] https://huggingface.co/stabilityai/stable-diffusion-3.5-large

[3] https://huggingface.co/black-forest-labs/FLUX.1-dev

[4] https://huggingface.co/Qwen/Qwen Image

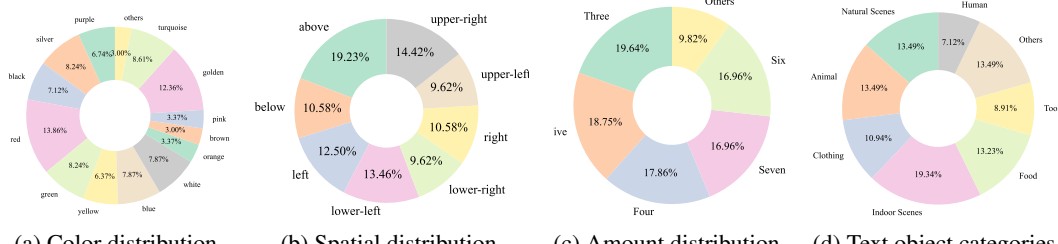

(a) Color distribution.  (b) Spatial distribution.  (c) Amount distribution.  (d) Text object categories.

Figure A1: Statistics of our constructed datasets for probing analysis. Each subfigure presents the distribution of prompts for a specific attribute: color, spatial relation, amount, and object category.

## D.3 Amount and CLIP Score Analysis for All Models

Due to space limitations, the main paper only presents the block-wise analysis results for the color and spatial relationship attributes on three models under the removing, disabling, and enhancing strategies, as well as the corresponding changes in DINOv2 similarity relative to the baseline. Here, we provide additional results, including the block-wise analysis for the amount attribute on all three models, and the overall CLIP score trends under the removing, disabling, and enhancing strategies for all models. The detailed results are shown in Fig. A2. Notably, the amount attribute exhibits a different sensitivity to the number of blocks compared to color and spatial attributes, and enhancement does not lead to a clear improvement, which may be attributed to the inherent limitations of MMDiT models in understanding quantity. The overall CLIP score trends are consistent with those of DINOv2 similarity, further validating the effectiveness and stability of our method.

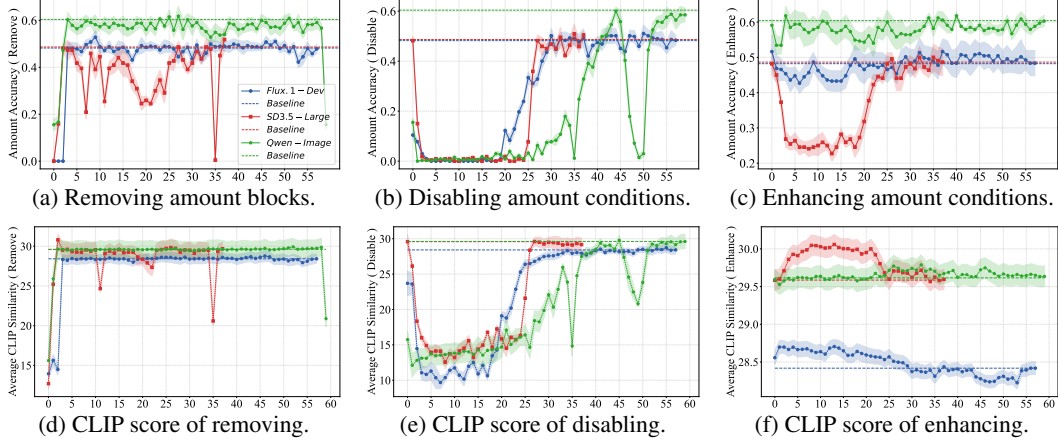

(a) Removing amount blocks.  (b) Disabling amount conditions.  (c) Enhancing amount conditions.

(d) CLIP score of removing.  (e) CLIP score of disabling.  (f) CLIP score of enhancing.

Figure A2: More detailed probing results on three MMDiT models: Stable Diffusion 3.5-Large, FLUX.1-Dev, and Qwen Image. Each subfigure presents the performance curves for a specific attribute (amount or overall CLIP score) under different probing strategies (removing, disabling, enhancing).

## D.4 Details of Evaluation for Probing Analysis

In probing analysis, we use the open-sourced Qwen2.5-VL 72B[5] (Bai et al., 2025) model for color and spatial relationship evaluation. We design specific systematic prompts to guide the model in accurately assessing whether the generated images align with the intended attributes in the text prompts. The detailed prompts for color and spatial relationship evaluation are provided in following colored boxes.

---

[5]https://github.com/QwenLM/Qwen2.5-VL

**Color Evaluation Prompt**

You are given an image, its caption, and a set of objects with their expected colors.
Your task: 1).For each object: check if the color in the caption matches the actual color in the image. 2).If the color matches, return "Yes". If the color does not match or the object is not visible, return "No".
Rules:

- Output ONLY a single valid JSON object.
- The JSON keys must be exactly the provided object names.
- The values must be strictly "Yes" or "No".
- Do not generate any other words.
- Do not add explanations, extra text, or formatting outside the JSON.

Example:

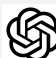 *Color Evaluation System Prompt ......*

*{bicycle: blue, wheels:yellow, wall: gray}* 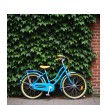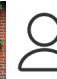

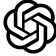 *{bicycle: yes, wheels:yes, wall:no}*

**Spatial Evaluation Prompt**

You are given an image, its caption, and a question about the spatial relationship between two objects in the image.
Your task: 1).Check whether the spatial relationship described in the question can be confirmed from the image.2).If the relationship is clearly visible and correct, return "Yes".3).If the relationship is not correct, cannot be seen, or the objects are unclear, return "No".
Rules:

- Output ONLY a single string. The value must be strictly "Yes" or "No".
- Do not generate any other words.
- Do not add explanations, extra text, or formatting outside the answer.

Example:

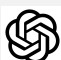 *Spatial Evaluation System Prompt ......*

*Is the chair above the giant ice cream cone?* 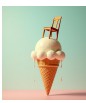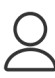

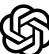 *Yes*

For amount evaluation, we use CountGD[6] (Amini-Naieni et al., 2024), an open-world object counting model based on Grounding-DINO (Liu et al., 2024b). The detection confidence threshold is set to 0.5 for higher precision. Counting accuracy is computed as the proportion of images where the predicted count exactly matches the ground-truth specified in the prompt.

**Statistical Significance.** We acknowledge that both the LLM-based evaluation and CountGD have inherent limitations. LLMs may misinterpret visual details or be affected by biases in their training data, while CountGD may produce inaccurate counts for small, overlapping, or occluded objects. To address these issues, we conduct multiple experimental runs with different random seeds and report averaged results, thereby reducing the impact of individual evaluation errors. To further ensure the validity and robustness of our conclusions, we additionally employ alternative evaluation methods that are independent of the primary approaches. This cross-validation helps to mitigate the influence of dataset bias and evaluation inaccuracies on our experimental findings.

---

[6]https://github.com/niki-amini-naieni/CountGD

# E    DETAILED SETUP OF EVALUATION RESULTS

## E.1    INFERENCE DETAILS

During evaluation of text-to-image generation, editing, and acceleration, we use the same hyperparameters in Tab. A1. The enhancement strength $\lambda$ is set to 1.5 by default. For the enhanced blocks, we select the block index in Tab. 1.

For all attributes except amount, we adopt a sentence-level approach, which has already demonstrated strong performance. For token-level enhancement, we construct the enhancement mask $M$ by passing the target phrases (*e.g.*, "two apples", "a person", "seven").

## E.2    EVALUATION BENCHMARKS

For text-to-image generation, we evaluate our method on the widely used T2I-CompBench++ (Huang et al., 2025) and GenEval (Ghosh et al., 2023) benchmarks. T2I-CompBench++ contains $8,000$ compositional prompts spanning color, spatial, 3D spatial, shape, texture, non-spatial relations, numeracy, and complex attributes. It extends the original benchmark (Huang et al., 2023) and introduces more challenging tasks (e.g., 3D spatial and numerical compositionality). We use all prompts and generate one sample per prompt. GenEval consists of 553 structured prompts targeting single-object, two-object, counting, color, position, and color–attribute binding. Each prompt is paired with four generated samples, and performance is computed with an automatic evaluation pipeline based on object detection, counting, and attribute classification, providing interpretable error types (e.g., missing objects, incorrect color, or miscount).

We evaluate image quality and text–image alignment using LAION Aesthetic v2 (Schuhmann, 2022) and HPSv2 (Wu et al., 2023). LAION Aesthetic v2 measures visual appeal, while HPSv2 evaluates prompt-image alignment relative to human judgments. As these metrics capture different aspects, we report both and supplement them with task-specific evaluations and human studies to ensure a comprehensive assessment.

## E.3    TOKEN-LEVEL ENHANCEMENT MASK LOCALIZATION

During inference, we record the multi-head self-attention of the concatenated features $Z_{in}$ at each MMDiT block and denoising step. To obtain a stable token-region corresponding, we aggregate the attention maps across all heads and denoising steps. Eventually, we get attention maps of shape $[N, H \times W, T]$, where $N$ is the number of MMDiT blocks, $H \times W$ is the spatial dimension of the image features, and $T$ is the number of text tokens. For visualization, we average all the MMDiT blocks' attention maps to get a single attention map of shape $[H \times W, T]$. We then normalize the attention maps along the spatial dimension and then resize them to the original image size. The visualization results are shown in Figure A3. We can see that 'dog' and 'cat' tokens have high attention values in the corresponding image regions, indicating that the token-level enhancement can effectively target specific areas in the image.

Based on the above analysis, we first tokenize the input prompt $\mathcal{P}$ using the same tokenizer as the MMDiT model, obtaining a sequence of token IDs $\mathbf{P} = [p_1, p_2, \ldots, p_N]$. Given a target phrase $\mathcal{Q}$ (e.g., "brown", "firetruck"), we tokenize it as $\mathbf{Q} = [q_1, q_2, \ldots, q_M]$. We then search for all subsequences in $\mathbf{P}$ that match $\mathbf{Q}$. The starting indices of these matches are collected in the set $\mathcal{I} = \{i \mid (p_i, p_{i+1}, \ldots, p_{i+M-1}) = (q_1, q_2, \ldots, q_M)\}$. The mask $M$ is constructed with the following rule:

$$M_j = \begin{cases} 1, & \text{if } \mathcal{I} = \varnothing \text{ or } \exists (i, i+M-1) \in \mathcal{I} \text{ with } j \in [i, i+M-1], \\ 0, & \text{otherwise.} \end{cases}$$

Here, $M_j$ indicates whether the $j$-th token in the prompt should be enhanced. If not matched, the token-level enhancement defaults to sentence-level enhancement by setting all entries of $M$ to 1.

## E.4    EDITING DETAILS

Stable Flow (Avrahami et al., 2025) is adopted as the baseline, which selects vital blocks based on the perceptual similarity. However, its block selection is not task-specific and may be inaccurate for

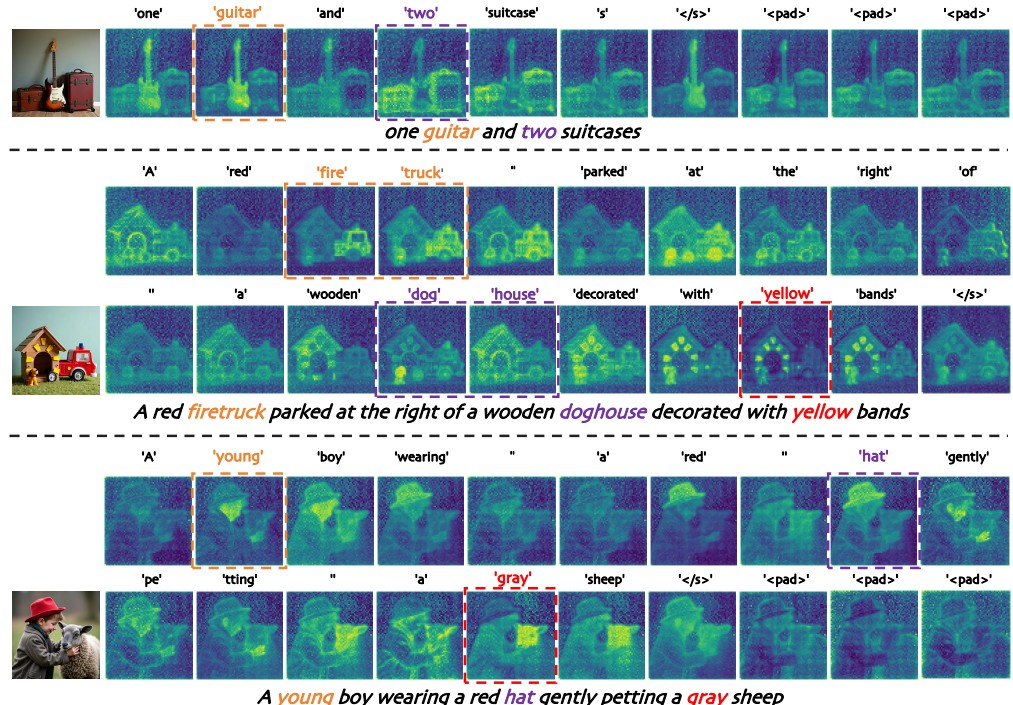

Figure A3: Visualization of token-level attention maps. Colored boxes indicate the target tokens in the prompt. The attention maps highlight the corresponding text tokens' indices in the image.

fine-grained editing, whereas our approach leverages probing analysis to identify blocks tailored to the editing task. Prior benchmarks lack coverage of quantity, attribute binding, and spatial relationship editing. To address this, we construct a new editing benchmark comprising $1,000$ images and corresponding editing prompts. Each source prompt is paired with four target prompts, covering object addition (with varying colors), background changes, color and lighting adjustments, shape and direction modifications, positional changes, quantity variations, and object actions. Evaluation is performed for each image–prompt pair using CLIP-image similarity and CLIP score to assess image quality and prompt adherence. Human evaluation details are provided in Section H.

### E.5 ACCELERATION DETAILS

Our method is conceptually related to TeaCache (Liu et al., 2025), which accelerates video diffusion by using timestep embeddings to estimate output differences and cache intermediate results selectively. In contrast, we skip blocks deemed irrelevant for the current editing task based on probing analysis. While TeaCache reduces redundancy across timesteps, our approach reduces computation across feature blocks, enabling acceleration without affecting editing quality.

For acceleration, we skip one-third of the CFG steps and remove three MMDiT blocks. Applying our method to both the FLUX baseline and TeaCache demonstrates significant speedup while maintaining comparable image quality. In each experiment, we randomly select $400$ prompts from T2I-CompBench++ and generate one sample per prompt. The CFG steps skipped are $[5, 10, 15, 20, 25, 30, 35, 40, 45, 50]$, and the removed MMDiT blocks are $[30, 40, 50]$. We repeat the experiments on both NVIDIA 4090 and H100 GPUs to verify the stability and robustness of our approach.

We present additional examples of accelerated generation in Fig. A4. The results in Tab. 5 and Fig. A4 demonstrate that our method can seamlessly integrate with TeaCache, achieving significant speedup while maintaining high-quality generation.

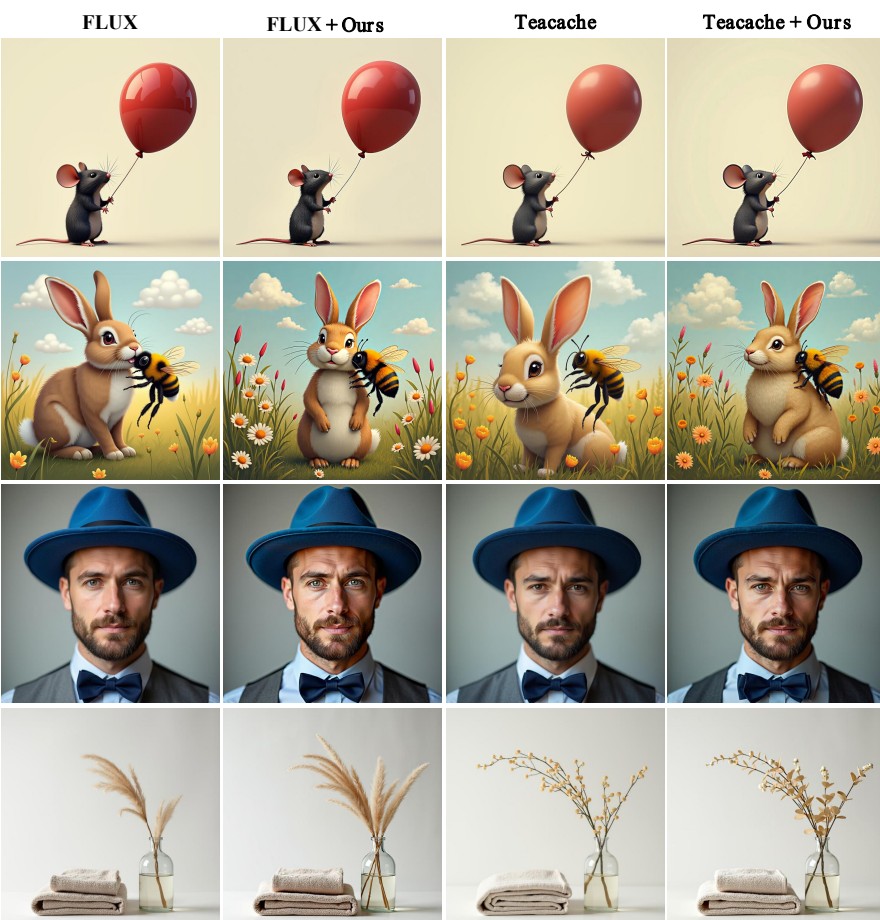

FLUX     FLUX + Ours     Teacache     Teacache + Ours

Figure A4: Examples of accelerated generation. Our method removes certain blocks, which may lead to different details in the generated image, yet the overall synthesis quality and textual semantics are consistent with the user prompts. The four prompts are (1) "a balloon on the right of a mouse", (2) "a rabbit hidden by a bee", (3) "a man in a blue and blur hat with a gray shirt and bowtie", (4) "a fabric towel and a glass vase".

# F    MORE QUANTITATIVE RESULTS

## F.1    ANALYSIS ON THE SELECTION OF ENHANCED BLOCKS

In the main paper, Tab. 6 only shows the results of color, shape, and spatial of different enhanced blocks and different strengthening scales. Here we provide the full results of T2I-CompBench++ in Tab. A2.

For the weakening experiments, we keep the same number and position of the enhanced blocks in the Tab. 1. All the metrics drop compared to the baseline and the 0.7 scale, even worse than the 0.9 scale, indicating that these vital blocks are indeed important for compositional generation. We also try to randomly select five blocks and all blocks to strengthen. Enhancing random five blocks can also improve the performance, but not as good as our selected blocks. This further validates the effectiveness of our probing analysis. Enhancing all blocks leads to a significant performance drop, which may be due to the excessive enhancement that distorts the original feature distribution and harms the generation quality.

Table A2: Full results of small $\lambda(l)$ enhancement and block selection experiments. All the experiments are conducted with FLUX on T2I-CompBench++.

| Methods | Attribute Binding | | | Object Relationship | | | Amount | Complex |
|---|---|---|---|---|---|---|---|---|
| | Color | Shape | Texture | 2D Spatial | 3D Spatial | Non-Spatial | | |
| FLUX | 0.7322 | 0.4908 | 0.6490 | 0.2935 | 0.3739 | 0.3044 | **0.5877** | 0.3597 |
| 0.7 | 0.3161 | 0.2653 | 0.2707 | 0.1100 | 0.1831 | 0.2890 | 0.3944 | 0.2630 |
| 0.9 | 0.6891 | 0.4395 | 0.5622 | 0.2611 | 0.3247 | 0.3029 | 0.5482 | 0.3419 |
| Random 5 blocks | 0.7624 | 0.5072 | 0.6492 | 0.3119 | 0.3797 | 0.3042 | 0.5842 | 0.3605 |
| All blocks | 0.2360 | 0.2736 | 0.2581 | 0.0495 | 0.1522 | 0.2928 | 0.2448 | 0.2233 |
| Ours | **0.7804** | **0.5482** | **0.6980** | **0.3280** | **0.3900** | **0.3054** | 0.5860 | **0.3691** |

## F.2    MORE ABLATION STUDIES ON SCALE SCHEMES

We study the impact of different enhancement scale schemes $\lambda(l)$, including uniform $(U(1.2, 1.8), U(1.8, 1.2))$, exponential $(Exp(1.6, 0.95))$, and fixed $(1.5)$ scales. As shown in Tab. A3, all schemes improve over the baseline, confirming the robustness of our selected enhancing blocks. The fixed scale is notably simple and achieves balanced performance across all dimensions, making it practical for general use. In contrast, other schemes have their own strengths: linearly increasing scales favor spatial performance, while decaying scales (e.g., $U(1.8, 1.2)$) further boost fine-grained attributes like color and shape. These findings indicate that both the magnitude and distribution of enhancement strength across blocks are important for compositional generation.

Table A3: Comparison of different scale schemes across blocks on T2I-CompBench++.

| Methods | Attribute Binding | | | Object Relationship | | | Amount | Complex |
|---|---|---|---|---|---|---|---|---|
| | Color | Shape | Texture | 2D Spatial | 3D Spatial | Non-Spatial | | |
| FLUX | 0.7322 | 0.4908 | 0.6490 | 0.2935 | 0.3739 | 0.3044 | **0.5877** | 0.3597 |
| U(1.2,1.8) | 0.7795 | 0.5438 | 0.6827 | **0.3311** | 0.3865 | 0.3051 | 0.5845 | 0.3685 |
| U(1.8,1.2) | **0.8035** | **0.5551** | 0.6940 | 0.3203 | 0.3893 | 0.3051 | 0.5654 | **0.3722** |
| Exp(1.6,0.95) | 0.7783 | 0.5507 | **0.7049** | 0.3296 | **0.3947** | **0.3060** | 0.5763 | 0.3710 |
| 1.5 (Fixed) | 0.7804 | 0.5482 | **0.6980** | 0.3280 | 0.3900 | 0.3054 | 0.5860 | 0.3691 |

### F.3 ORIGINAL DATA FOR ENHANCING SCALE AND BLOCK ABLATION

In Fig. 7, we present experiments conducted with a fixed enhancing scale ranging from 1.2 to 2.0, as well as experiments varying the number of enhancing blocks. In the main text, the vertical axes were normalized to emphasize consistent trends across different models and evaluation metrics. Here, we provide the original, unnormalized data for reference (see Tab. A4 and Tab. A5 for details). In addition, we include the corresponding results for SD3.5 under the same enhancing scale settings. It could be observed that the results consistently reflect identical conclusions with the main paper.

Table A4: Results of the enhancing scale experiments. Except the FLUX data presented in the main text, we also provide the corresponding results for SD3.5 under the same experimental settings.

| Methods/$\lambda(l)$ | | Baseline | 1.2 | 1.3 | 1.4 | 1.5 | 1.6 | 1.8 | 2.0 |
|---|---|---|---|---|---|---|---|---|---|
| FLUX | color | 0.7322 | 0.7776 | 0.7768 | 0.7775 | **0.7804** | 0.7708 | 0.7643 | 0.7415 |
| | shape | 0.4908 | 0.5389 | 0.5489 | 0.5381 | **0.5482** | 0.5449 | 0.5226 | 0.5093 |
| | spatial | 0.6603 | 0.6827 | 0.7244 | 0.6987 | **0.7340** | 0.7147 | 0.7083 | 0.6795 |
| SD3.5 | color | 0.7284 | 0.7992 | 0.8064 | **0.8072** | 0.8052 | 0.7874 | 0.7785 | 0.7608 |
| | shape | 0.5592 | 0.6432 | 0.6656 | 0.6642 | **0.6744** | 0.6659 | 0.6255 | 0.6048 |
| | spatial | 0.6418 | 0.6683 | 0.7596 | 0.7716 | 0.7885 | **0.7933** | 0.7740 | 0.7486 |

Table A5: Results of experiments varying the number of enhancing blocks on FLUX.

| Methods | Baseline | 1 | 3 | 5 | 7 | 9 |
|---|---|---|---|---|---|---|
| color | 0.7365 | 0.7533 | 0.7776 | **0.7863** | 0.7613 | 0.7073 |
| shape | 0.4720 | 0.5159 | 0.5181 | **0.5254** | 0.5201 | 0.4919 |
| spatial | 0.2985 | 0.3109 | 0.3152 | **0.3327** | 0.2841 | 0.2581 |

### F.4 MORE RESULTS ABOUT TOKEN-LEVEL ENHANCEMENT

As shown in Table A6, token-level enhancement generally provides more precise guidance compared to sentence-level enhancement, leading to consistent but modest improvements across amount-related attributes. The gains, however, remain limited, which can be attributed to the intrinsic weakness of current diffusion models in numerical reasoning and counting.

Table A6: Token-level vs. sentence-level enhancement on amount-related attributes.

| Methods | Amount(T2I-CompBench++) | Count(GenEval) |
|---|---|---|
| SD3.5 | 0.5969 | 0.6344 |
| ours(sentence) | 0.5929 | 0.6125 |
| ours(token) | **0.6088** | **0.6375** |
| FLUX | 0.5877 | 0.6375 |
| ours(sentence) | 0.5860 | 0.6000 |
| ours(token) | **0.6091** | **0.6438** |
| Qwen Image | 0.7406 | 0.8562 |
| ours(sentence) | 0.7359 | 0.8275 |
| ours(token) | **0.7616** | **0.8594** |

## F.5 DETAILED ACCELERATION RESULTS

In the main paper Tab. 5, we evaluate the results of our method for inference acceleration by removing less critical blocks. Here we provide more detailed results in Tab. A7, including the time cost on both NVIDIA 4090 and H100 GPUs, and the image quality metrics (HPSV2, LaionAesthetic V2, CLIP-Text) on different models. We test two baseline models: the original FLUX.1-Dev and the TeaCache-optimized version. For each baseline, we apply our method with different CFG step skipping and block removal strategies.

The CFG steps skipped are $seq(5, 50, 10)$, $seq(5, 55, 5)$, and $seq(6, 58, 3)$, and the removed MMDiT blocks are $[30, 40, 50]$, where $seq(a, b, c)$ denotes the arithmetic sequence starting from $a$ to $b$ with step $c$. We can see that skipping more CFG steps leads to faster inference without significantly affecting image quality. We finally choose to skip CFG $seq(6, 58, 3)$ steps and remove $[30, 40, 50]$ blocks as the default setting for a good trade-off between speed and quality.

Table A7: Details of acceleration

| Method | Time(4090)↓ | Time(H100)↓ | HPSV2↑ | LaionAes V2↑ | CLIP-Text↑ |
|---|---|---|---|---|---|
| FLUX | 36.7889 | 13.0876 | **29.0533** | 6.1903 | **26.9986** |
| skip CFG $seq(5, 50, 10)$ | 35.5859 | 12.6414 | 29.0395 | **6.2081** | 26.9761 |
| skip CFG $seq(5, 55, 5)$ | 33.6433 | 11.9846 | 28.7259 | 6.1340 | 26.8841 |
| skip CFG $seq(6, 58, 3)$ | **31.6931** | **11.3010** | 28.8408 | 6.1034 | 26.7460 |
| Ours | 33.3387 | 11.8734 | 28.9212 | 6.1874 | 26.7807 |
| Teacache | 26.6187 | 9.6125 | 28.8951 | 6.2067 | 26.8346 |
| skip CFG $seq(5, 50, 10)$ | 26.1385 | 9.4414 | 28.9054 | 6.1983 | 26.8646 |
| skip CFG $seq(5, 55, 5)$ | 25.2994 | 9.1565 | 28.8822 | 6.1973 | **26.8984** |
| skip CFG $seq(6, 58, 3)$ | **24.5276** | **8.8804** | 28.8647 | 6.1801 | 26.7946 |
| Ours | 24.9992 | 9.0743 | **28.9481** | **6.2256** | 26.7722 |

# G  MORE QUALITATIVE RESULTS

## G.1  MORE SD3.5 RESULTS

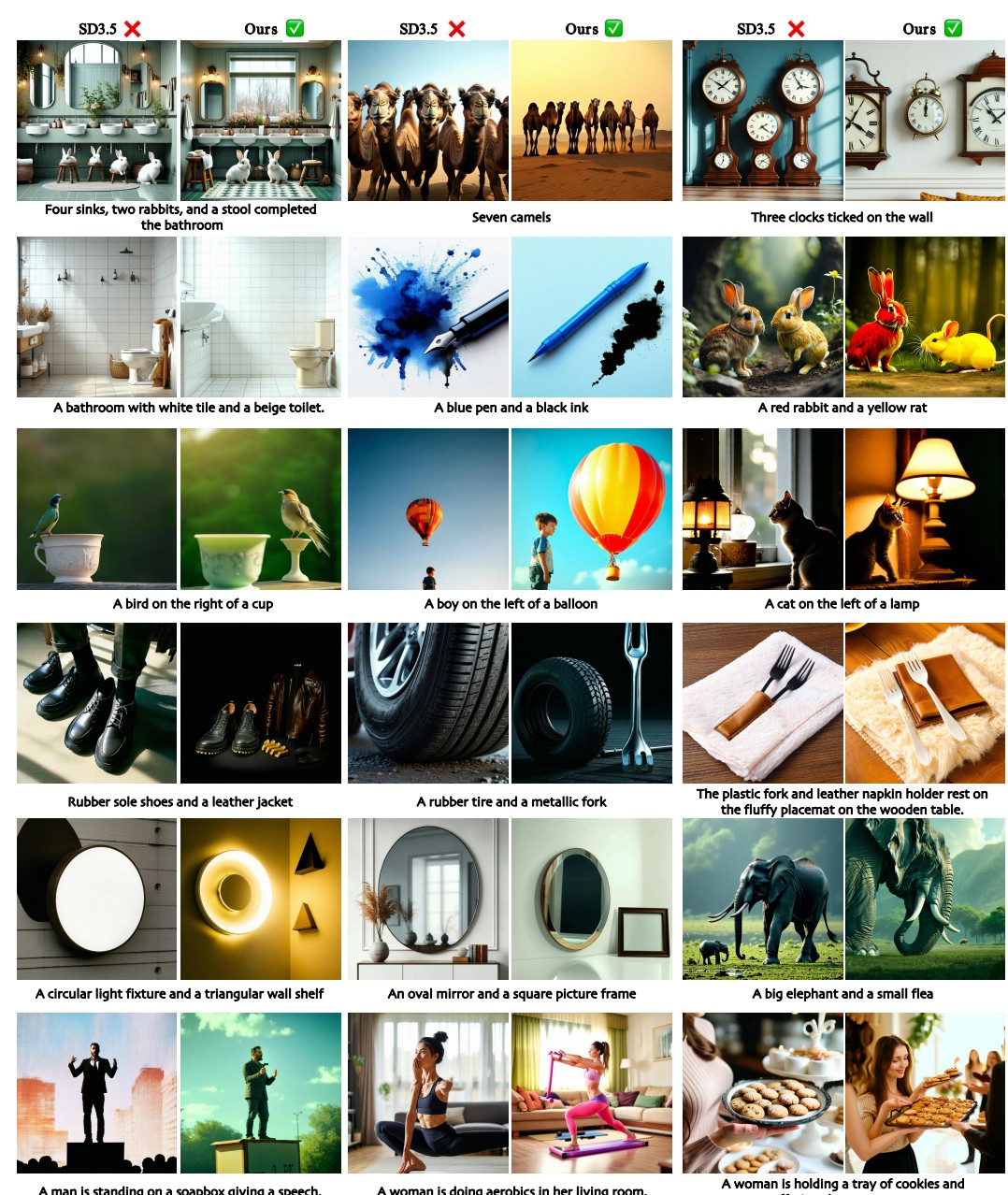

Figure A5: More qualitative results of SD3.5 and our method, covering aspects such as amount, color, spatial arrangement, texture, shape, and non-CLIP attributes. Our method consistently demonstrates better text alignment.

## G.2 MORE FLUX RESULTS

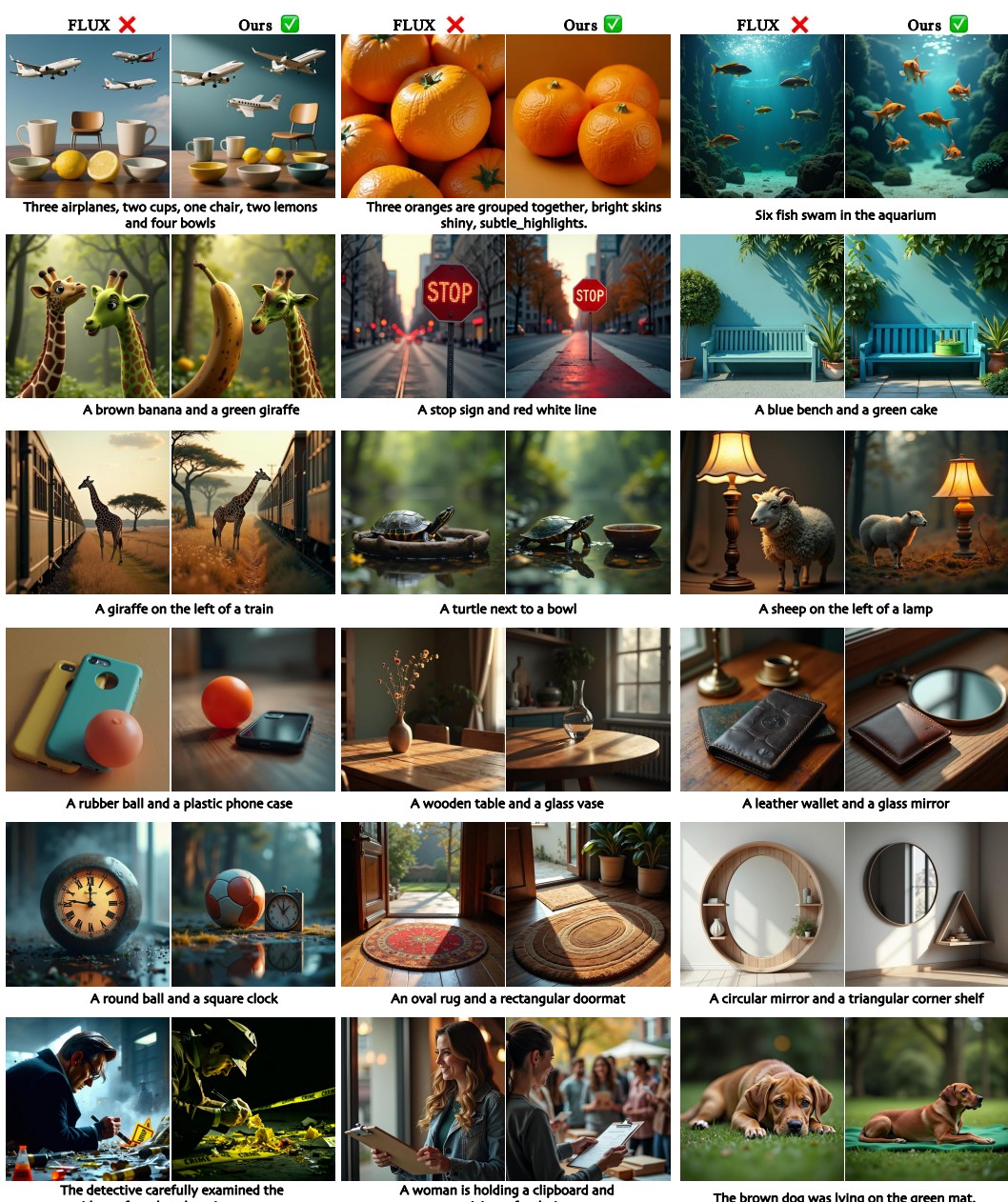

Figure A6: More qualitative results of FLUX and our method. Our approach achieves better text-image alignment and, in some cases, improved aesthetics over the baseline.

### G.3 MORE QWEN IMAGE RESULTS

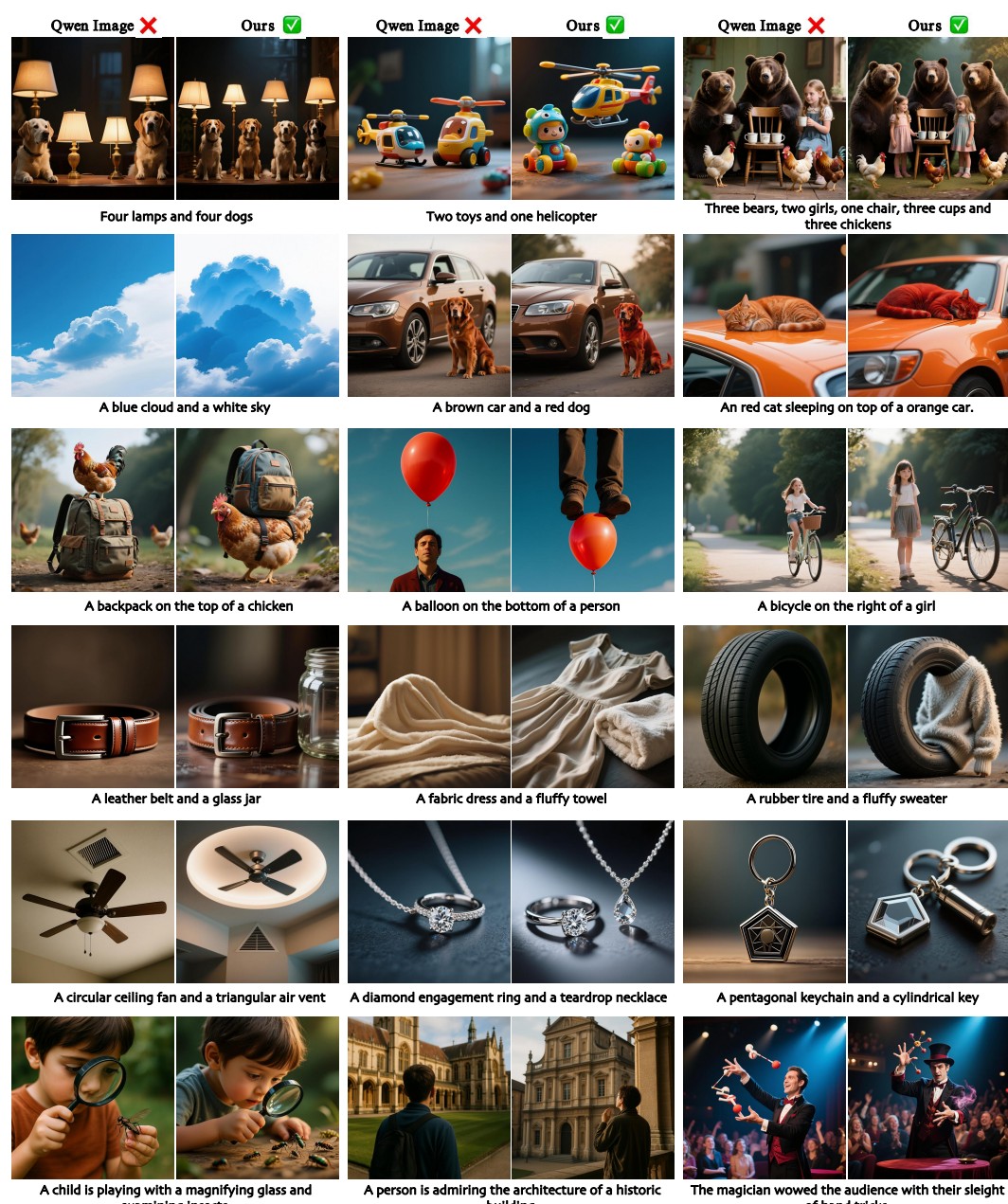

Figure A7: More qualitative results of Qwen Image and our method. Although the baseline fails on "A blue cloud and a white sky", our method succeeds.

## G.4 More Editing Results

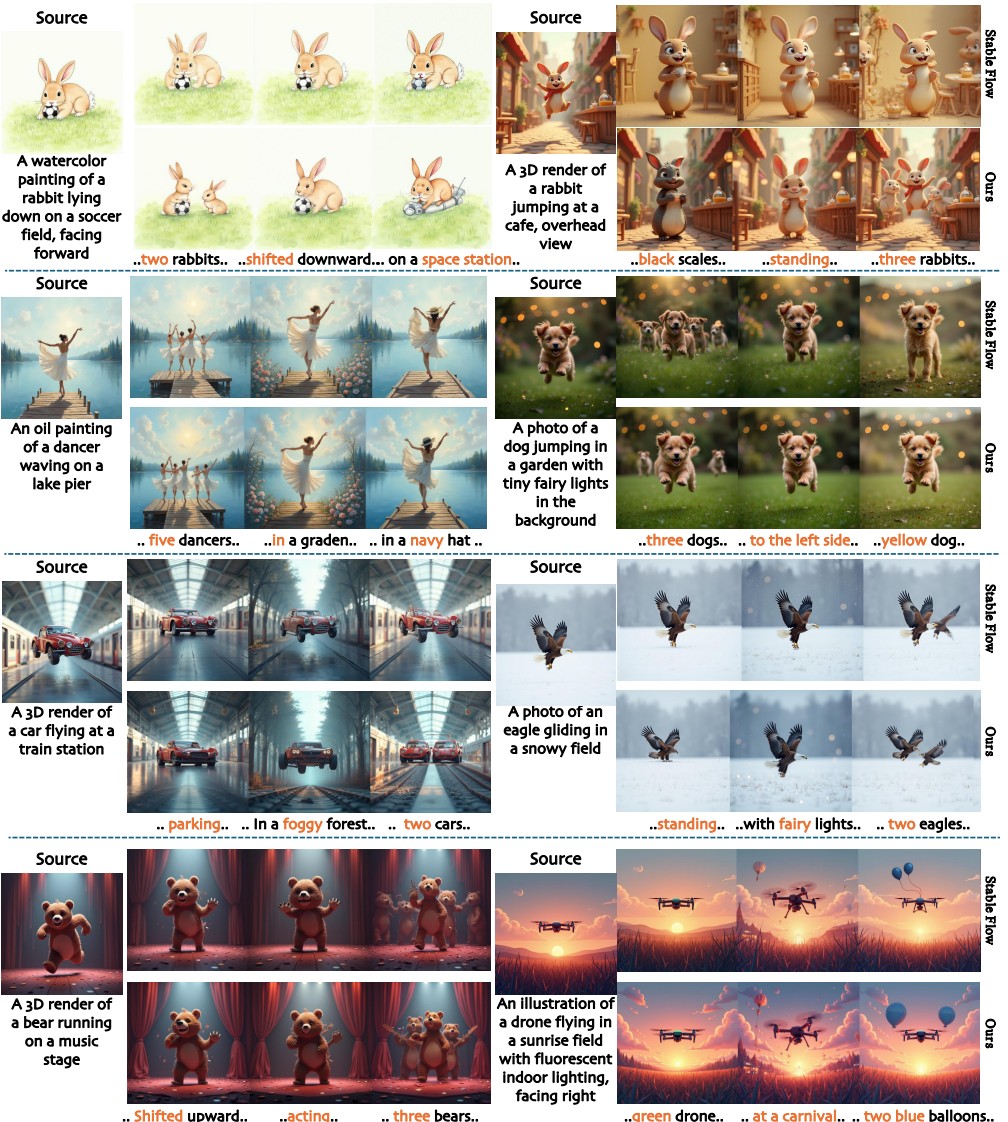

Figure A8: More editing examples on FLUX with our method and StableFlow. Our approach particularly surpasses StableFlow in quantity while maintaining high fidelity and strong text-image alignment.

## G.5 Failure Cases

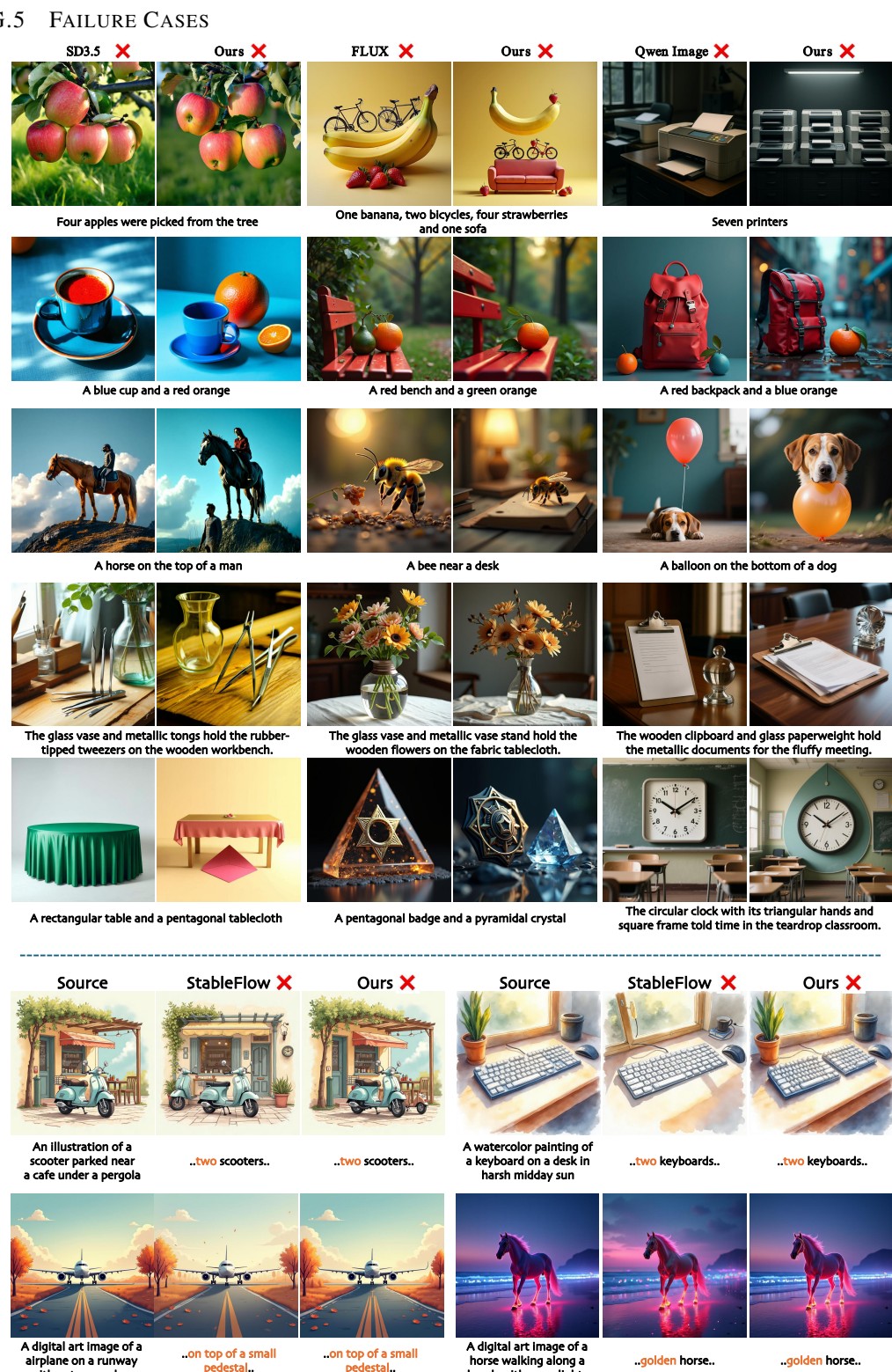

Figure A9: Failure cases of generation and editing. SD3.5 sometimes misidentifies attributes, confusing colors and objects. For rare real-world cases, our method may produce correct attributes but also hallucinations, e.g., a dog missing its body. In editing, hard cases mainly involve amount, reflecting the model's limited counting ability.

## G.6 ALL-BLOCK SHOWCASES OF PROBING ANALYSIS

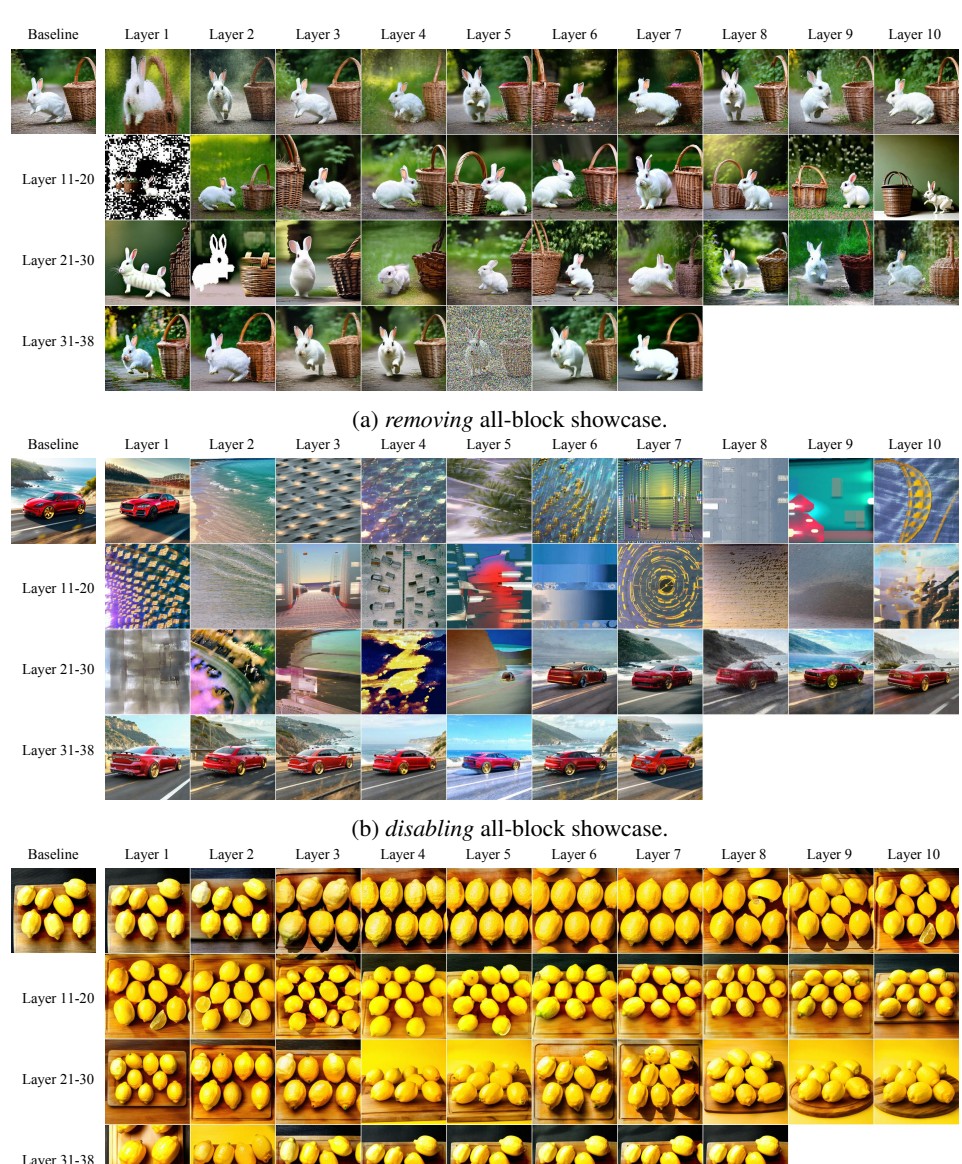

(a) *removing* all-block showcase.

(b) *disabling* all-block showcase.

(c) *enhancing* all-block showcase.

Figure A10: All-block showcases of probing analysis on Stable Diffusion 3.5-large. From top to bottom, the prompts are: (a)."A white rabbit is hopping to the right of a brown basket.",(b)."A red car with golden rims speeds along a coastal road.", (c)."Seven lemons are arranged on a wooden cutting board, skins textured, color bright."

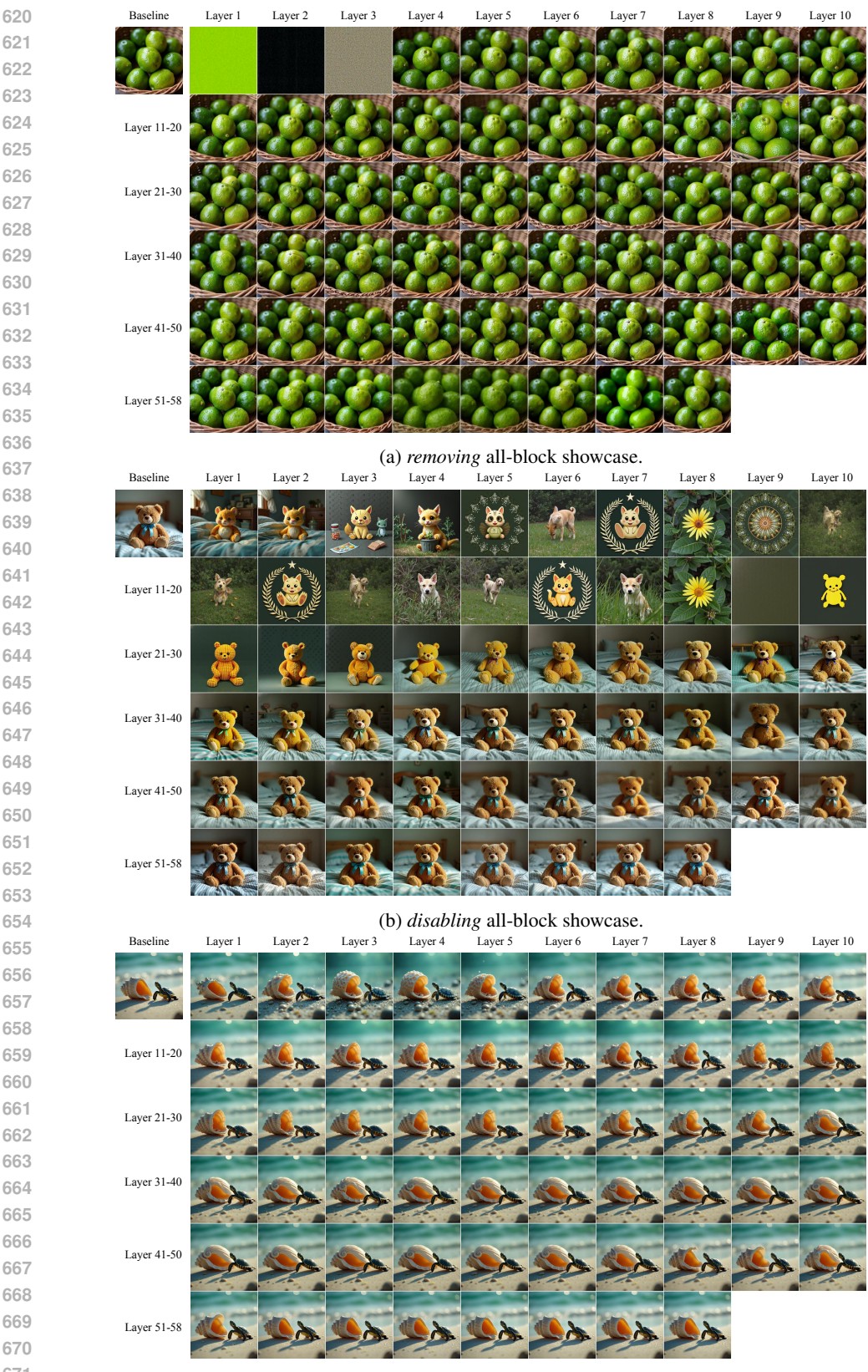

(a) *removing* all-block showcase.

(b) *disabling* all-block showcase.

(c) *enhancing* all-block showcase.

Figure A11: All-block showcases of probing analysis on FLUX.1-Dev. From top to bottom, the prompts are: (a)."Eight green limes rest in a basket, shiny skins and small droplets of water visible.",(b)."A brown teddy bear with a blue ribbon sits on a child's bed with striped sheets.", (c)."A small turtle moves to the right of a seashell."

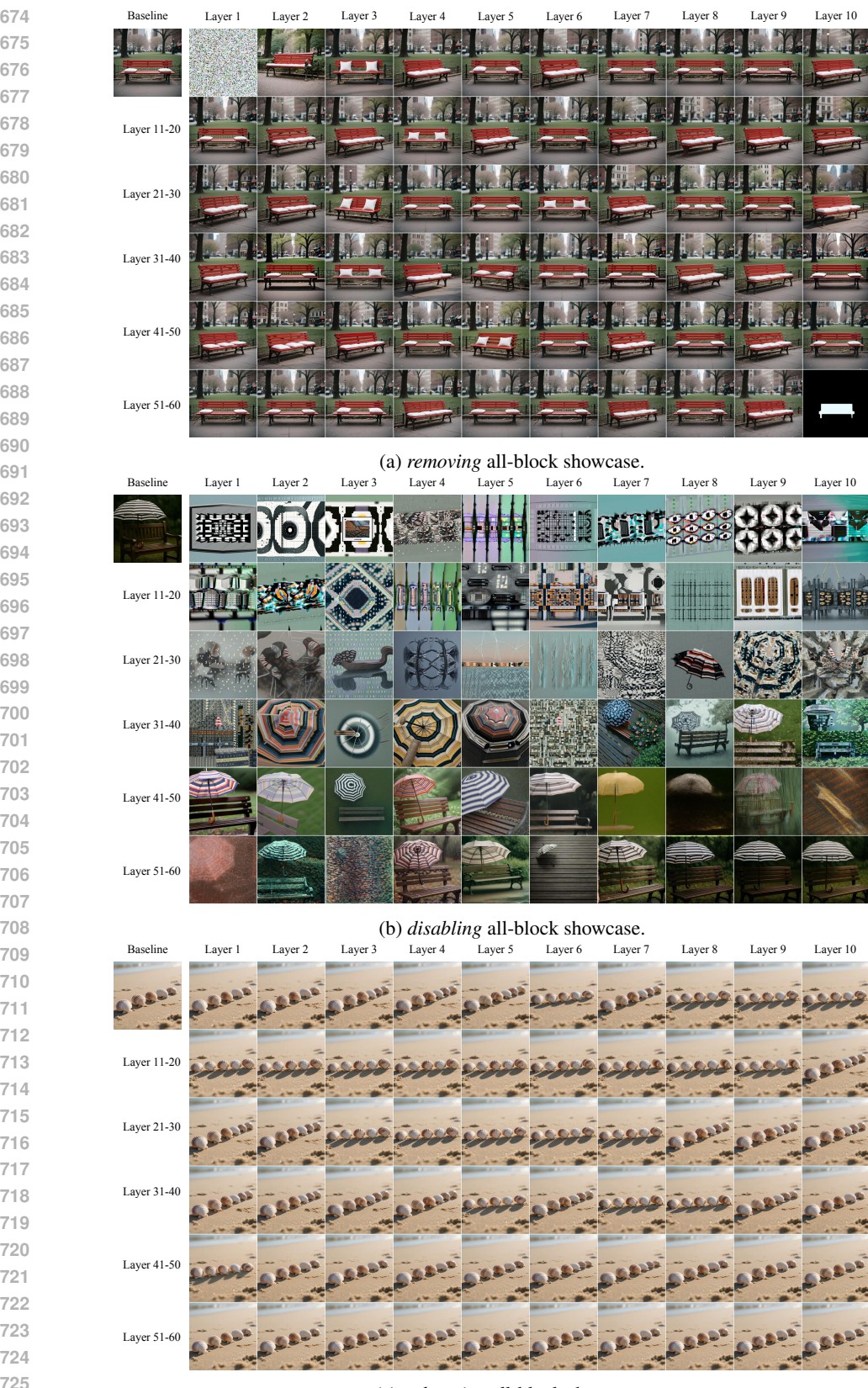

(a) *removing* all-block showcase.

(b) *disabling* all-block showcase.

(c) *enhancing* all-block showcase.

Figure A12: All-block showcases of probing analysis on Qwen Image. From top to bottom, the prompts are: (a)."A red bench with white cushions stands in a quiet city park.",(b)."A striped umbrella is positioned at the top-left of a wooden bench.", (c)."Five seashells lie in a row on sand, each detailed with ridges and soft reflections."

## H  HUMAN EVALUATION DETAILS

We conduct human evaluations for both generation and editing tasks. For text-to-image generation, we sample 100 prompts from T2I-Compbench++ and generate results using FLUX.1-Dev and our method. Participants are provided with the prompt and the two generated images in random order, and answer the questions in Fig. A13a. The human preference score is defined as

$$\text{Human Preference} = \tfrac{1}{4}\text{Alignment} + \tfrac{1}{4}\text{Aesthetic} + \tfrac{1}{2}\text{Overall}.$$

For image editing, we randomly select 100 samples from the editing dataset and apply our method and the baseline Avrahami et al. (2025) with identical prompts. Three participants are recruited and, given the original image, the editing prompt, and two edited results in random order, they answer the questions in Fig. A13b. The final preference score is computed as

$$\text{Human Preference} = \tfrac{1}{4}\text{Alignment} + \tfrac{1}{4}\text{Preservation} + \tfrac{1}{2}\text{Overall}.$$

The corresponding quantitative results are summarized in Tab. A8 and Tab. A9.

Table A8: Human evaluation results for generation.

| Methods | Alignment | Aesthetic | Overall | Human Preference |
|---------|-----------|-----------|---------|------------------|
| FLUX    | 118       | 134       | 109     | 39.17%           |
| + Ours  | 182       | 166       | 191     | 60.83%           |

Table A9: Human evaluation results for editing.

| Methods | Alignment | Preservation | Overall | Human Preference |
|---------|-----------|--------------|---------|------------------|
| StableFlow | 124    | 136          | 115     | 40.83%           |
| Our Method | 176    | 164          | 185     | 59.17%           |

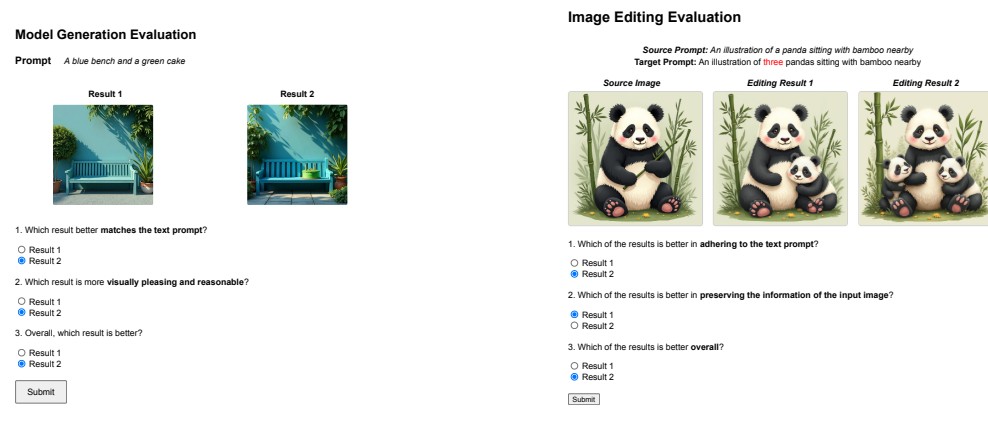

(a) Interface for generation evaluation.                    (b) Interface for editing evaluation

Figure A13: Human evaluation interfaces for (a) generation and (b) editing. Participants are asked three questions corresponding to the given prompts and images.

