# OpenReview forum: "Revisiting Block-wise Interactions of MMDiT for Training-free Improved Synthesis"
_ICLR.cc/2026/Conference — ICLR 2026 Conference Withdrawn Submission_

### Official Review · Reviewer_xhmo · 2025-10-24

**Soundness:** 2
**Presentation:** 2
**Contribution:** 3
**Rating:** 2
**Confidence:** 4

**Summary:**

In this paper, the authors investigate the role of different layers within the diffusion transformer by skipping a block, removing attention to the textual representations in a block, or enhancing the attention to the textual tokens in a block. The authors construct a dataset of 333 prompts with GPT-5 with attributes like color and shape, and use these interventions to find out what blocks are more important for each task. Later, the authors use the selected layers to improve generation, by enhancing the attention to textual tokens in the found layers.

**Strengths:**

The authors evaluate 3 common open source T2I models, analyze their behaviour and use the insights from their analysis to improve generation. The authors analyse the specific role of different layers, which is novel.

**Weaknesses:**

The analysis of removing specific blocks was already presented in [1] on general prompts. While the main novelty of the authors in this paper in my opinion is to make a better localization, with specific layers attributed to specific tasks (like color and shape), the authors only use 333 prompts to do so. Since it’s done automatically, I would expect scale here. Moreover, the random baseline of selecting 5 random layers (table 6) looks to work well, almost as well as in the case you choose the layers. Moreover, the final results are evaluated on CLIP score to evaluate the performance of the improvement, while CLIP is considered poor for this kind of tasks [2]. Finally, no std is provided for the main results. A few more minor comments in the questions section.

[1] Stable Flow: Vital Layers for Training-Free Image Editing (Avrahami 2025)
[2] WHEN AND WHY VISION-LANGUAGE MODELS BEHAVE LIKE BAGS-OF-WORDS, AND WHAT TO DO
ABOUT IT? (Yuksekgonul, 2022)

**Questions:**

1. Section 2.2 “Disabling textual conditions is more disruptive than removing specific blocks” - removing specific blocks includes disabling textual conditions and disabling self-attention in the image. The fact that you find that disabling textual conditions is more disruptive is probably an artifact of the self-attention without condition. I don't see the value of this observation, please explain why you think this finding is interesting.
2. Line 160: “we construct a challenging prompt dataset with GPT-5, comprising 333 diverse and difficult prompts across three attributes: color, amount, and spatial relationships” - How many of each category? How are these prompts constructed?
3. How are blocks selected for each category? Do you use a threshold? How many prompts are used for each selection?
4. The evaluation in the main results is done with CLIP score, known to be poor for such a nuanced task. Why not to use a better metric?
5. Human preference - what is the agreement between human annotators?
6. No STD in main tables (2,3)
7. Acceleration improvement is not 2x in Table 5, while in the line 311 you say “Notably, when classifier-free guidance (CFG) is enabled during inference, we can achieve a ×2inference time acceleration on both the conditional and unconditional predictions.” why do the results differ so much?
8. What blocks are selected for skipping?
9. Line 428: “We can derive from the table that enhancing randomly selected or all blocks underperforms enhancing dedicated blocks identified from our analysis“ but Table 6: Random 5 blocks achieve almost the same improvement that you get.
10. Minor: Fig 1 text in last row goes on the image. Fig 3: g,h,i are missing “)” on y axis

---

### Official Review · Reviewer_QnGY · 2025-10-24

**Soundness:** 2
**Presentation:** 3
**Contribution:** 2
**Rating:** 2
**Confidence:** 4

**Summary:**

This work explores how block-wise manipulations—specifically removing blocks, disabling text conditioning, or enhancing it—impact the output of SOTA image generative models. The authors demonstrate that this analysis enables training-free applications, such as accelerating the inference process by removing less critical blocks and improving synthesis quality by enhancing text representations.

**Strengths:**

- The paper presents an interesting use case for accelerating inference that only minimally impacts the quality of generated images, which may be applicable in situations.
- The evaluation is performed on SOTA models, ensuring the relevance of the findings.
- The writing is clear and the methodology is well-explained.

**Weaknesses:**

- Sample size (3) for human evaluation is very small, authors should consider doing more extended study.
- Similar methodology of exploring importance of specific blocks in diffusion models is already introduced in a numbers of works, such as [1], [2], [3] or [4]. Moreover these works explore the importance in light of many attributes not explored in this work, such as style, objects, safety, text. This would could largely benefit from extending the number of explored attributes.
- Related works mentioned above use similar methodology of studying importance of blocks and thus findings from "removing blocks" and "disabling text conditioning" are not novel.
- The results for the Qwen Image model (plots 3a, d, g) show that as the number of blocks increases, removing single blocks results in almost no distinguishable change in accuracy. This suggests the proposed analysis may become less informative as models continue to grow.
- The observation that disabling textual conditions is more disruptive than removing blocks (Sec 2.2) seems largely attributable to the experimental setup. Overwriting text conditioning with zeros in a specific block causes this modified representation to propagate through all subsequent layers. It is therefore an expected outcome that this would cause a greater disruption than simply omitting one block's computation, and the direct comparison may be misleading.
- The enhancing conditions approach yields unstable results, improving DINOv2 similarity for color and spatial attributes but degrading it for the amount attribute. This raises practical concerns, as it would be difficult to determine a priori if the method will provide a consistent performance boost for a given prompt.

[1] Basu, Samyadeep, et al. "On mechanistic knowledge localization in text-to-image generative models." _Forty-first International Conference on Machine Learning_. 2024.

[2] Zarei, Arman, et al. "Localizing Knowledge in Diffusion Transformers." _arXiv preprint arXiv:2505.18832_ (2025).

[3] Staniszewski, Łukasz, et al. "Precise Parameter Localization for Textual Generation in Diffusion Models." _The Thirteenth International Conference on Learning Representations_.

[4] Avrahami, Omri, et al. "Stable flow: Vital layers for training-free image editing." _Proceedings of the Computer Vision and Pattern Recognition Conference_. 2025.

**Questions:**

- The DINOv2 similarity metric indicates that the "enhancing conditions" approach introduces a trade-off: while accuracy may increase for certain attributes, perceptual similarity decreases. Could you investigate this trade-off more thoroughly and discuss its implications?
- How does the enhancing conditioning approach perform for other attributes not covered in the paper, such as style, object fidelity, or composition?
- Given that the impact of removing single blocks diminishes in larger models, have you considered investigating alternative probing strategies, such as removing contiguous groups of blocks, which might better isolate where specific attributes are synthesized?

---

### Official Review · Reviewer_UY8D · 2025-11-01

**Soundness:** 3
**Presentation:** 3
**Contribution:** 2
**Rating:** 4
**Confidence:** 4

**Summary:**

The paper studies block-wise interactions in MMDiT-style diffusion transformers by systematically (i) removing blocks, (ii) disabling text conditions, and (iii) enhancing text hidden states in selected blocks. Using these insights, the authors propose training-free tweaks for text-image alignment, instruction-based editing (with Stable-Flow-style parallel generation and self-attention injection), and inference acceleration by skipping low-impact blocks. Main reported gains appear on T2I-CompBench++ and GenEval, while quality metrics (HPSv2, Aesthetics) are maintained. The acceleration scheme skips blocks and can combine with TeaCache.

**Strengths:**

- The analysis of remove/disable/enhance triad leads to intuitive takeaways: earlier blocks capture semantics (e.g., color, spatial relations) while later blocks refine details; disabling text is more damaging than dropping a block; and targeted enhancement can help.

- The tweak is training-free and easy to use -- it is just element-wise scaling of text hidden states in selected blocks (optionally with token-level masks, Eq. 5), with no architectural changes.

- The method improves Shape/Texture/Color on T2I benchmarks and delivers competitive editing results against Stable Flow, supported by both metrics and human preference.

**Weaknesses:**

- The work appear to be incremental wrt Stable Flow. In addition, its editing setup also closely follows Stable Flow—parallel generation from source/target prompts and self-attention injection—with the main novelty being attribute-aware block selection and text amplification. The approach hinges on identifying “pivotal” blocks per attribute (color/amount/spatial). It is unclear how to scale this to more open-ended instructions without an impractically broad probing stage; the paper’s core analyses and reported wins emphasize those three structured categories.

- Token-level masks (Eq. 5) are mainly demonstrated for the editing setup and amount-related attributes; broader trials for general T2I or diverse instructions are not shown in the main text (details are largely in supplementary token-localization and small amount-focused tables).

- The paper positions itself as MMDiT-focused, but key techniques (sequence-level text scaling after text–image concatenation) also apply to DiT variants. The work does not analyze how multi-stream vs single-stream designs (e.g., early multi-stream/late single-stream mixes as reported for FLUX) affect cross-modal Q/K/V/MLP interactions; the mechanism-level picture remains surface-level, even though the paper assumes joint attention over concatenated tokens.

**Questions:**

- How would you automate block selection for open-ended instructions beyond color/amount/spatial, without a costly per-task probing pass? Any proxy predictors that generalize?

- For multi-stream vs single-stream hybrids (e.g., FLUX), can you provide evidence on where the gains come from in terms of Q/K/V/MLP pathway interactions? Does the “pivotal block” distribution shift compared to fully single-stream MMDiT?

---

### Note · Authors · 2025-11-13

I have read and agree with the venue's withdrawal policy on behalf of myself and my co-authors.